# Assistive Teaching of Motor Control Tasks to Humans

**Megha Srivastava**
Stanford University
megha@cs.stanford.edu

**Erdem Bıyık**
UC Berkeley
ebiyik@berkeley.edu

**Suvir Mirchandani**
Stanford University
suvir@cs.stanford.edu

**Noah D. Goodman**
Stanford University
ngoodman@stanford.edu

**Dorsa Sadigh**
Stanford University
dorsa@cs.stanford.edu

## Abstract

Recent works on shared autonomy and assistive-AI technologies, such as assistive robot teleoperation, seek to model and help human users with limited ability in a fixed task. However, these approaches often fail to account for humans' ability to adapt and eventually learn how to execute a control task themselves. Furthermore, in applications where it may be desirable for a human to intervene, these methods may inhibit their ability to learn how to succeed with full self-control. In this paper, we focus on the problem of *assistive teaching* of motor control tasks such as parking a car or landing an aircraft. Despite their ubiquitous role in humans' daily activities and occupations, motor tasks are rarely taught in a uniform way due to their high complexity and variance. We propose an AI-assisted teaching algorithm that leverages skill discovery methods from reinforcement learning (RL) to (i) break down any motor control task into teachable skills, (ii) construct novel drill sequences, and (iii) individualize curricula to students with different capabilities. Through an extensive mix of synthetic and user studies on two motor control tasks—parking a car with a joystick and writing characters from the Balinese alphabet—we show that assisted teaching with skills improves student performance by around 40% compared to practicing full trajectories without skills, and practicing with individualized drills can result in up to 25% further improvement.[1]

## 1 Introduction

Imagine a novice human is tasked with operating a machine with challenging or unfamiliar controls, such as landing an aircraft or parking an oversized vehicle with novel transmission. A fully autonomous form of assistance, such as a self-driving car, would aid this user by entirely replacing their control, seeking interventions only when necessary. Shared autonomous systems, such as those used in assistive robot teleoperation [1, 2, 3], would seek to model the human's goals and capabilities, and provide assistance in the form of corrections [4] or simplified control spaces [3]. However, neither form of assistance would actually help the user *learn* how to successfully operate the machine, and may even serve as a crutch that prevents their ability to ever perform the task independently.

Although humans can learn a variety of complex motor control tasks (e.g., driving a car, or operating surgical robots), they often rely on the assistance of specialized teachers. Receiving fine-grained instruction from these specialized teachers can often be non-uniform, costly, and limited by their availability. In this work, we are interested in a more accessible and efficient approach: we wish to develop an AI-assisted teaching algorithm capable of teaching motor control tasks in an individualized manner. While several works have leveraged AI techniques to aid instruction in traditional education tasks such as arithmetic [5, 6, 7] and foreign language learning [8, 9], motor control tasks introduce

---

[1]Our source code is available at https://github.com/Stanford-ILIAD/teaching.

several unique challenges, such as the high complexity of the input and output space for motor control tasks (e.g., controlling a high degree of freedom robotic arm with a 6 degrees of freedom joystick or the continuous output space of trajectories in driving), as well as the need for generalization across different scenarios in these high-dimensional and complex spaces.

To address these challenges, our insight is to utilize compositional motor skills for teaching: skills are more manageable to teach, and can be used to construct novel compositional drills for an individual. We take inspiration from how specialized human teachers teach motor control tasks. For example, a piano teacher may assign a student structured exercises (e.g., musical scales or arpeggios) to build up their general technique. Further, they may guide the student to break down complex musical phrases into more manageable chunks, repeatedly practice those, and then compose them together [10].

However, such fine-grained teaching often requires access to a specialized teacher capable of identifying skills and creating individualized drills based on a student's unique expertise. This is not only expensive, but can be highly suboptimal for many motor control tasks where a teacher may not have full visibility into the student's various inputs for a task (e.g. how much the student pushed the brake pedal), but only observability of the final behavior of the system (e.g. how close the vehicle is to the curbside), leading to less meaningful feedback. Can we leverage the rich action data provided by students learning motor control tasks to develop more efficient and reliable individualized instruction?

Our key insight is to leverage automated skill discovery methods from the RL literature to break down motor control tasks performed by an expert into *teachable skills*. We then identify which of the expert's skills are skills that the student is struggling with, and construct *novel drills* that focus on those particular skills to *individualize* their curriculum. Our contributions include: 1) Developing an algorithm to identify which skills of a motor control task a student struggles with, 2) Automatically creating individualized drills based on students' expertise in various skills, and 3) Empirically validating the helpfulness of our AI-assisted teaching framework in two different motor control tasks: controlling a simulated vehicle to park with an unintuitive joystick, and writing Balinese words using a computer mouse or trackpad. Our results show that assisted teaching with skills improve student performance by around 40% compared to practicing full trajectories without skills, and practicing with individualized drills can result in up to 25% further improvement.

## 2   Related work

**Skill discovery in Reinforcement Learning.** A large body of work has studied how to leverage behavioral primitives, or skills in RL. Hierarchical RL approaches [11, 12] use skills as temporal abstractions to accelerate learning [13, 14, 15]. However, hand-designing skills can be difficult, especially in continuous spaces. Several works proposed methods to automatically discover skills [16, 17, 18, 19, 20]. Works in unsupervised skill discovery derive task-agnostic skills directly from an environment, without a reward function [21, 22, 23, 24, 25, 26]. Skills can also be learned from a set of tasks [27] or demonstrations [28, 29]; e.g., CompILE [28] learns to segment and reconstruct given trajectories, using learned segment encodings to represent skills. Rather than using discovered skills to aid the learning of automated agents (e.g., via RL or planning [23, 24, 25]), we investigate the novel application of discovered skills to teaching *humans*.

**Assistive human-AI systems.** Human-AI collaborative systems have the potential to augment a human's individual capabilities. In the paradigm of *shared autonomy* [30], an AI assistant modulates a user's observation [31] or input on tasks such as robot teleoperation [1, 2, 3]. Unlike our work, these works generally assume the human's policy is fixed, and do not directly aim for a human to learn from the AI agent [4]. Bragg et al. [32] improve both collaborative performance and user learning in a shared autonomy framework, but they rely on simulated users in their evaluation. We present a technique to support human learning to improve individual proficiency, and rigorously evaluate it on real users.

**AI-assisted teaching.** A distinct line of work has focused on developing algorithms for helping humans learn. *Knowledge tracing* attempts to model a student's knowledge state in terms of skills relevant to a set of tasks, yet have generally required manual skill annotations [33] or large-scale student data [34]. *Instructional sequencing* uses a model of student learning to select a sequence of instructional activities to maximize learning, but these approaches have generally focused on generating curricula for domains like language learning, arithmetic, and logic [35]. Similarly, *machine teaching* [36] explores the problem of selecting optimal training data for a student with a known

model of learning, and has been effective in discrete domains with structured knowledge such as mathematics and visual classification [37].

We build on the idea that a student's proficiency can be represented in terms of different skills, and an AI assistant can adapt a curriculum accordingly. However, to the best of our knowledge, we are the first to explore how to teach motor control tasks informed by student actions. This setup has unique challenges: it is less obvious how to delineate skills, how to handle student feedback (in the form of continuous trajectories versus binary correct/incorrect responses), and how to measure a student's proficiency at a skill. By leveraging advances from skill discovery in RL, we do not require expert annotation of skills nor large-scale student data.

**Motor learning in humans.** *Motor skill learning* in psychology refers to the acquisition of skilled actions as a result of practice or experience [38, 39]. Our approach is informed by several theories in motor learning and specifically the common insights that human motor control operates hierarchically, and that direct practice of individual skills generally aids skill learning [38, 40].

## 3 Formulation

In this section, we formalize the problem of *Student-Aware AI-Assisted Teaching*. We go over preliminaries and formalize skill discovery, and then define the problem statement leading to our 3-step approach of teaching complex motor control tasks: (i) distilling tasks into underlying skills, (ii) identifying the student's proficiency, and (iii) creating novel and individualized curricula.

### 3.1 Preliminaries

We formalize the target motor control task as a standard Markov decision process (MDP) $\langle \mathcal{S}, \mathcal{A}, \rho, f, R, T \rangle$ with finite horizon $T$, where $\rho$ is the initial state distribution and $f$ is a deterministic transition function. In our formulation, $R$ is a random variable for reward functions such that each $r \sim R$ is a function $r : \mathcal{S} \times \mathcal{A} \to \mathbb{R}$. Given an initial state $s_0 \sim \rho$ and a reward function $r \sim R$ (both of which are revealed at the beginning of the task), the goal is to maximize the cumulative reward over the horizon. We call any $(s_0, r)$ pair a scenario $\xi$ drawn from a set of all possible scenarios $\Xi$. We assume access to a set of expert demonstrations, e.g., demonstrations drawn from a pre-trained RL agent that optimizes the cumulative reward for any given scenario.

An inexperienced human would lack the expertise to perform optimally in a given scenario $\xi$. Our goal is to develop teaching algorithms that leverage expert demonstrations to improve student performance. For example, we want a human unfamiliar with Balinese to learn to write any Balinese word, which would correspond to one $r$. For this, we develop an algorithm that uses skill discovery methods to automatically create individualized drills based on the discovered skills and the student performance.

**Skill discovery.** Humans often break down complex tasks into relevant skills (i.e., compositional action sequences that lead to high performance)—when learning how to write, we first master how to write letters, and then compose them to write words and sentences. We define a skill as a latent variable $m$ that lies in some finite discrete latent space $\mathcal{M}$. Each skill is an embedding that corresponds to an action sequence $\Lambda = (a_1, a_2, \dots, a_{|\Lambda|})$ optimal at some state $s \in \mathcal{S}$ under some reward $r \sim R$:

$$\exists s \in \mathcal{S}, \exists r : \quad P(R = r) > 0, \quad \Lambda = (a_1^*, a_2^*, \dots, a_{|\Lambda|}^*),$$
$$(a_1^*, a_2^*, \dots, a_{|\Lambda|}^*, \dots) \in \underset{a_1, a_2, \dots, a_{|\Lambda|}, \dots}{\arg\max} \left( r(s, a_1) + r(f(s, a_1), a_2) + \dots \right), \quad (1)$$

and $|\Lambda| \in [H_{\min}, H_{\max}]$, where $H_{\min}$ and $H_{\max} \ll T$ are hyperparameters.

In many tasks, it is difficult to enumerate all possible skills, and it is often not clear when a skill starts and ends in a trajectory. In this work, we use skill discovery algorithms to automatically extract skills relevant to a task, and then teach the student those generated skills. While our teaching method is agnostic to the skill discovery algorithm, we use CompILE [28] as it learns skills directly from expert trajectories. Our formulation assumes access to expert trajectories, so it would be unnecessary to learn an expert policy in parallel with the skill discovery process, as several other methods do.

We use the trained CompILE as a function: SKILLEXTRACTOR($\tau$) where $\tau$ is a trajectory. It outputs a sequence of skills $M^\tau = (m_1^\tau, m_2^\tau, \dots)$ and boundaries $B^\tau = \left( b_0^\tau, b_1^\tau, \dots, b_{N_{\text{seg}}}^\tau \right)$ such that $0 = b_0^\tau < b_1^\tau < \dots < b_{N_{\text{seg}}}^\tau = T$, dividing $\tau$ into $N_{\text{seg}}$ *segments*. Actions in $\tau$ between steps $[b_{j-1}^\tau, b_j^\tau)$ correspond to skill $m_j^\tau \in \mathcal{M}$. Effectively, CompILE learns to divide trajectories into segments, and cluster segments with similar actions into skills. We refer to [28] and the Appendix for more details.

## 3.2 Problem statement

Having formalized SKILLEXTRACTOR, which, given a trajectory $\tau$, outputs $M^\tau$ and $B^\tau$, we now present our problem statement for AI-assisted teaching of motor control tasks.

**What should the student do?** Given a set of expert demonstrations in scenarios $\Xi^e \subseteq \Xi$, where each $\xi \in \Xi^e$ is sampled from $(\rho, R)$, we would ideally teach the student to eventually have a policy $\pi^*: \mathcal{S} \to \mathcal{A}$ that minimizes the difference between their behavior and the expert demonstrations:

$$\pi^* = \arg\min_\pi \sum\nolimits_{\xi \in \Xi} P(\xi) \mathcal{L}(\tau_\xi^e, \tau_\xi^\pi) \approx \arg\min_\pi \sum\nolimits_{\xi \in \Xi^e} \mathcal{L}(\tau_\xi^e, \tau_\xi^\pi), \qquad (2)$$

where $\tau_\xi^\pi$ denotes the student trajectory performed by policy $\pi$ in scenario $\xi$. Similarly, $\tau_\xi^e$ denotes the expert trajectory in $\xi$.[2] $\mathcal{L}$ is a task-dependent loss function. While it is a straightforward goal, it is not clear how to teach a human to optimize this objective to achieve expert-like trajectories for any $\xi$.

**What should the teacher do?** We can formulate the teaching problem as a partially observable MDP (POMDP), where the state of this POMDP is the student's policy (initially $\pi_0$), and the teacher's actions guide the student to the policy $\pi^*$. The teacher's observations would be student behavior expressed by their trajectories, or $\tau_\xi^{\pi_k}$ for some $\xi$ for state $\pi_k$ at teaching round $k$. The teacher can then select actions (e.g. practice scenarios) in this POMDP. A teaching action at round $k$ moves the student from $\pi_k$ to $\pi_{k+1}$,

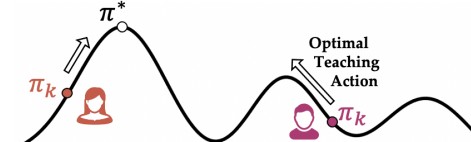

Figure 1: An optimal teacher should select a teaching action (e.g. which practice scenario to provide) based on each student's individual policy to help guide them towards the expert policy $\pi^*$.

giving the teacher a reward of $\sum_{\xi \in \Xi} P(\xi) \left( \mathcal{L}(\tau_\xi^e, \tau_\xi^{\pi_k}) - \mathcal{L}(\tau_\xi^e, \tau_\xi^{\pi_{k+1}}) \right)$, i.e., the decrease in the loss value according to Eq. (2). Thus, the teacher should take the optimal teaching actions that maximize its cumulative reward over teaching rounds, whose global optimum would move the student to $\pi^*$ (see Fig. 1).

Unfortunately, we lack computational models of human learning necessary for making the POMDP's state transition function (i.e., how a student updates its policy from $\pi_k$ to $\pi_{k+1}$ for any teaching action the teacher could take) tractable for motor control tasks. Recent works exploring data-driven frameworks for knowledge tracing [34] and RL-based education [6] usually require large amounts of data for simpler problems, and are thus not scalable to motor control tasks with large trajectory spaces.

**Approach overview.** Our insight is that once we identify important skills of a motor control task, an AI assisted teaching framework can efficiently leverage the skills to teach the task. The SKILLEXTRACTOR function allows us to identify the skills used in a given trajectory. Applying it to all expert demonstrations in scenarios $\Xi^e \subseteq \Xi$ gives us a set of skills relevant in the task. Say we label the expert demonstration $\tau_\xi^e$ for each scenario $\xi \in \Xi^e$ with skills and their intervals: $(M_\xi^e, B_\xi^e) = \text{SKILLEXTRACTOR}(\tau_\xi^e)$. We would then want the student to perform each relevant skill as close as possible to the expert to solve the following optimization:

$$\pi^* = \arg\min_\pi \sum\nolimits_{\xi \in \Xi} P(\xi) \sum\nolimits_{m \in M_\xi^e} \mathcal{L}_m(\tau_\xi^e, \tau_\xi^\pi) \approx \arg\min_\pi \sum\nolimits_{\xi \in \Xi^e} \sum\nolimits_{m \in M_\xi^e} \mathcal{L}_m(\tau_\xi^e, \tau_\xi^\pi), \quad (3)$$

where $\mathcal{L}_m$ is some loss function that compares expert and student trajectories in terms of the skill $m$. The intuition behind this objective is two-fold: (i) we want the student to perform the same set of skills as the expert for any $\xi \in \Xi$ (this is why we sum over $m \in M_\xi^e$), and (ii) we want them to perform each skill as close as possible to how the expert performs it. The underlying assumption is that student will gain expertise if they perform the same skills in the most similar way to the expert.

Overall, (3) is a simpler objective as the space of action sequences for each skill is much smaller than the space of trajectories or scenarios. We can now try to teach a small number of skills to the student instead of unrealistically trying to match them with the expert in the large trajectory space.

Only teaching popular skills of a task may suffer from the problem that the student does not learn how to compose different skills to achieve the target task. We thus propose to create novel *drills*, which consist of repetitions of multiple skills in a single trajectory. Drills help the students (i) learn how to connect action sequences for different skills, and (ii) develop muscle memory, which is

---

[2]For concision, we assume a single expert demonstration $\tau_\xi^e$ and at most a single student trajectory $\tau_\xi^\pi$ for any $\xi \in \Xi^e$. Without this assumption, we would take expectations over expert and student trajectories for each $\xi$.

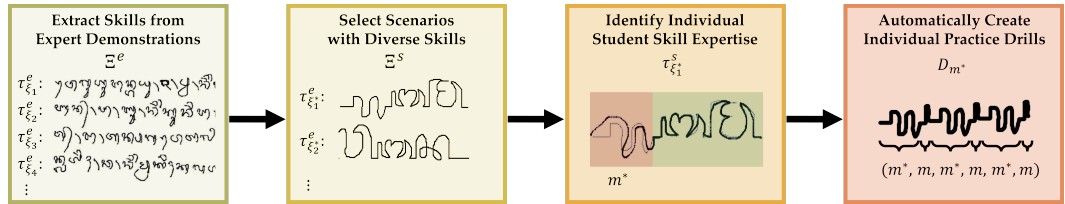

Figure 2: **Overview of our approach.** We train a CompILE model over expert demonstrations (e.g., handwritten Balinese text) for our SKILLEXTRACTOR function, which we use to select a diverse set of scenarios (e.g. words) covering many skills. After a student provides trajectories for these scenarios, we use SKILLEXTRACTOR again to identify their individual expertise for each skill, which informs the set of drills we provide them to practice.

crucial in motor control tasks, for the action sequences of each skill. The generated drills can also be personalized based on what skills each student is struggling with. For instance, as in Fig. 1, two different students (in green and blue) might have different policies with different skill sets. Therefore, a teacher should generate *individualized drills* informed by the student's initial level of expertise.

To efficiently solve (3), we propose our student-aware teaching approach that follows three steps: **1) Diverse scenario selection:** Identify scenarios that yield a high diversity of popular skills when an expert controls the system, **2) Expertise estimation:** Have the student control the system under these scenarios to estimate their expertise in different skills, and **3) Individualized drill generation:** Based on the estimates, create individualized drills to have the student practice the skills that they are struggling with. In the subsequent section, we describe our procedure for each of these steps.

## 4 Student-aware assisted teaching

**1) Diverse scenario selection.** We select diverse scenarios that cover many different skills in order flexibly estimate students' expertise and individualize our teaching strategy. For example, in English, we may have expert demonstrations of writing the words (scenarios) "expert", "person", "skills", from which we would select "person" to cover more diverse letter strokes (skills). Formally, we treat the problem of choosing scenarios with diverse skills as a maximum set coverage problem, which we solve using a simple greedy algorithm (see Algorithm 1), outputting a set of scenarios that led the expert to demonstrate a variety of skills.

---

**Algorithm 1** Diverse Scenario Selection

**Input:** Skill labels $M_\xi^e$ for each $\xi \in \Xi^e$
**Input:** Number of scenarios to be selected $N^s$
1: $\Xi^s, M_{\text{covered}} \leftarrow \emptyset, \emptyset$
2: **for** $i = 1, 2, \ldots, N^s$ **do**
3: $\quad \xi^* \leftarrow \arg\max_{\xi \in \Xi^e \setminus \Xi^s} |M_{\text{covered}} \cup M_\xi^e|$
4: $\quad \Xi^s \leftarrow \Xi^s \cup \{\xi^*\}$
5: $\quad M_{\text{covered}} \leftarrow M_{\text{covered}} \cup M_\xi^e$
6: **return** $\Xi^s$

---

**2) Identifying individual expertise in skills.** To efficiently teach complex tasks, we seek to estimate student's expertise in different skills, such as identifying which strokes they struggle with when writing Balinese words. Requiring a diverse set of skills, the scenarios $\Xi^s$ we select enable us to infer student's expertise efficiently: we ask the student to perform the task in those scenarios, and based on their data, construct a labels vector $E$ that quantifies the student's expertise $E_m$ for each skill $m \in \mathcal{M}$.

The discrepancy between skills used by the student and the expert provides a useful signal to estimate the student's expertise. Intuitively, we expect the student to demonstrate similar skills

---

**Algorithm 2** Individual Expertise Identification

**Input:** Selected scenarios $\Xi^s \subseteq \Xi^e$
**Input:** Skill labels $M_\xi^e$ for each $\xi \in \Xi^e$
**Input:** Student demonstration $\tau_\xi^s$ for each $\xi \in \Xi^s$
1: $E_m \leftarrow 0$ for $\forall m \in \mathcal{M}$
2: **for** $\xi \in \Xi^s$ **do**
3: $\quad M_\xi^s, B_\xi^s \leftarrow$ SKILLEXTRACTOR$(\tau_\xi^s)$
4: $\quad j \leftarrow 0$
5: $\quad$ **for** $m \in M_\xi^e$ **do**
6: $\quad\quad$ **if** $m \notin M_\xi^s$ **then**
7: $\quad\quad\quad E_m \leftarrow E_m + r_\xi(\tau_\xi^s)/j$
8: $\quad\quad j \leftarrow j + 1$
9: **return** $E$

---

to the expert if they are proficient in those skills, while, otherwise, they might rely on a simpler or incorrect set of skills. For example, a student driver who lacks the expertise of reverse driving might attempt to park a vehicle by only driving forward. We therefore utilize the expert demonstrations in the same scenarios the student performed the task by extracting the skills the student used via SKILLEXTRACTOR and comparing them with the expert skills in those scenarios. The student's expertise in a skill $m$ is then: $E_m = -\sum_{\xi \in \Xi^s} \Delta_m(\tau_\xi^e, \tau_\xi^s)$, where $\tau_\xi^s$ is the student trajectory in $\xi$, and $\Delta_m$ is a function that computes the discrepancy between the two trajectories in terms of $m$.

In our implementation, if a skill is used only by the expert, we assume the student did not attempt or failed that skill. We increase the discrepancy based on how far in the trajectory they failed. We assign higher penalty to failures in the beginning of the scenario for two reasons: 1) Skills relevant early in the task are often more important: other skills would not even be needed without first achieving these skills. 2) Failing at early skills might mean the student never had a chance or a need to perform the later skills due to compounding errors, so our uncertainty about the later skills is higher. Thus, we decide to be more conservative about the later skills. Specifically, we use:

$$\Delta_m(\tau_\xi^e, \tau_\xi^s) = \begin{cases} -r_\xi(\tau_\xi^s)/j & \text{if } m \notin M_\xi^s \text{ and } m_j^{\tau_\xi^e} = m, \\ 0 & \text{otherwise.} \end{cases} \tag{4}$$

Here $r_\xi$ is the reward function under scenario $\xi$, which we assume to be non-positive.[3] We increase the discrepancy if a skill is only performed by the expert, and we discount this value over time ($j$ in Eq. (4)). We also scale discrepancy with $r_\xi(\tau_\xi^s)$ so that student trajectories with lower reward will be penalized more in terms of expertise. Alg. 2 presents the pseudocode for this step. In applications where the reward function is not readily available, one could design the skill-discrepancy function $\Delta_m$ independent of the rewards by using a divergence metric between expert and student trajectories.

**3) Creating individualized drills.** Referring to Eq. (3), a student should master the skills they struggle with to get better at the overall task. Therefore, the teacher should first identify those skills, and then have the student practice them. For example, if a teacher thinks a student who attempted to write the word "person" struggled with writing the letter "p", they could provide the word "puppet" as a *drill* for the student. Using the expertise vector $E$ for a student, which is the output of Alg. 2, we can select the skills $m^*$ with low $E_{m^*}$ as the skills that the student needs to practice.

Given a skill $m^*$, the student should ideally practice it in the context of skills that frequently go together with $m^*$, rather than in isolation or with arbitrary skills. Our approach uses the expert demonstrations to identify frequent skill sequences containing skill $m^*$. We create *drills* consisting of these sequences that enable us to have the student practice (i) the target skill $m^*$, and (ii) how skill $m^*$ connects with other skills that co-exist in various scenarios. Each drill repeats such a sequence $N_{\text{rep}}$ times. A natural approach is to iterate over all $n$-grams, borrowing terminology from natural language processing, of skills from the expert demonstrations. Within each $n$-gram, we check if $m^*$ exists to identify the skills that often co-occur with $m^*$. Mathematically, *frequency* of an $n$-gram of skills $x$ is (where $\mathbb{I}$ is an indicator for the particular $n$-gram $x$):

---

**Algorithm 3** Individualized Drill Creation

**Input:** Individual expertise vector $E$
**Input:** Number of skills to create drills for $N_{\text{target}}$
**Input:** Hyperparameters $n, N_{\text{drills}}, N_{\text{rep}}$
**Input:** Expert demonstration $\tau_\xi^e$ for each $\xi \in \Xi^e$
**Input:** Skill labels $M_\xi^e$ for each $\xi \in \Xi^e$
1: Compute $f(x)$ for all $n$-grams $x$   ▷ Eq. (5)
2: **for** $m^* \leftarrow \arg \text{set}_{N_{\text{target}}} \min_m E_m$ **do**   ▷ see †
3:     $X \leftarrow \arg \text{set}_{N_{\text{drills}}} \max_{x \ni m^*} f(x)$   ▷ see †
4:     $D_{m^*} \leftarrow \bigcup_{x \in X} \bigoplus_{i=1}^{N_{\text{rep}}} \bigoplus_{m \in x} \tau_m^e$   ▷ see ‡
5: **return** $D$

---

† $\arg \text{set}_n \min_m E_m$ returns the set of $n$ distinct indices that individually minimize $E_m$.
‡ $\tau_m^e$ is the segment of an expert demonstration from interval $[b_{j-1}^{\tau^e}, b_j^{\tau^e})$ such that $m_j^{\tau^e} = m$. The operator $\oplus$ concatenates the actions of trajectory segments.

$$f(x) = \sum_{\xi \in \Xi^e} \sum_{i=1}^{|M_\xi^e| - n + 1} \mathbb{I}\left(x = (m_i^{\tau_\xi^e}, m_{i+1}^{\tau_\xi^e}, \ldots, m_{i+n-1}^{\tau_\xi^e})\right), \tag{5}$$

We can then ask the student to practice the $N_{\text{drills}}$ $n$-grams involving the target skill $m^*$ with the highest frequency.

Next, we formalize a drill as a repetition of the skills in the $n$-grams. This ensures the student gets enough practice of skill $m^*$ with the most common skill sequence containing $m^*$. We then project this constructed skill sequence back to the trajectory space by again using the expert demonstrations. While different segments of the expert demonstrations may correspond to the same skill, we randomly select one of those segments to project back to the trajectory space.[4] Alg. 3 presents the pseudocode. It returns a set of drills $D_{m^*}$ for every target skill $m^*$. Though these drills can be used in several ways, e.g., explaining them to a student, here we let the student observe and then imitate each drill trajectory.

---

[3]This is a mild assumption, since we can always subtract a constant offset from any finite reward function.
[4]We set the initial state of each drill as the initial state of the first trajectory segment in the drill. Since the transition function $f$ is deterministic, the initial state and action sequence specify a full trajectory.

# 5 Experiments

**Environments.** We consider two motor control tasks: parking a car in simulation with a joystick controller (PARKING), and writing Balinese characters (WRITING) using a computer mouse [5].

WRITING: We introduce a novel task of writing Balinese character sequences[6] from the Omniglot dataset [41], which contains action sequences for 1623 characters spanning 50 alphabets provided by crowdworkers from Amazon Mechanical Turk. We sample 5 Balinese characters, introduce a connector character "-", and create goal sequences of up to 8 characters. An agent in this environment operates in 2D $(x, y)$ continuous state and action space, and receives reward corresponding to the degree of overlap between the agent's and the gold trajectory. We train CompILE with 1000 random sequences based on trajectories provided by human contributors to the Omniglot dataset.

PARKING: We use the Parking environment from HighwayEnv [42], a goal-based continuous control task where an agent must park a car in a specified parking spot by controlling its heading and acceleration (2D action space). The 6D state space corresponds to the car's position, velocity, and heading and at each time step, and the agent receives a reward representing how close the current state is to the goal state. Scenarios differ across initial heading angles and goal parking spots, resulting in a diverse set of skills needed. We train CompILE with rollouts across random scenarios from an optimal RL agent trained for $10^6$ epochs using the StableBaselines [43] implementation of Soft Actor-Critic.

For both tasks, reward is negative with an optimal value of 0, but reported with a positive offset. Further details, as well as necessary pre-processing steps for both environments and all hyperparameters used for training CompILE models as part of SKILLEXTRACTOR, are in the Appendix.

**Drill Creation.** We create one drill for each latent skill identified by SKILLEXTRACTOR for both PARKING ($n = 3, N_{rep} = 1, N_{target} = 7, N_{drills} = 1$) and WRITING ($n = 2, N_{rep} = 3, N_{target} = 8, N_{drills} = 1$) tasks. Fig. 6 shows examples of these drills.

**User Study.** We design web applications for our user study for both environments. For PARKING, students see the environment rendered on the interface, and control the car's steering and acceleration with a 2D joystick. For WRITING, students use their computer mouse or trackpad to write on an HTML5 canvas, with the goal sequence of Balinese characters displayed underneath. Students in both environments follow a sequence of pre-test scenarios, practice sessions (including skills or drill-based practice), and evaluation scenarios. We recruit students through the Prolific platform, and include further information about pay rate, average experiment times, and task interfaces in the Appendix.

## 5.1 Results

**Does individualization help synthetic students?** To demonstrate our full approach, we first show results for synthetic "students" in the PARKING environment. We train two different synthetic students using standard behavior cloning on goal-conditioned $(s, a)$ tuples from expert demonstrations:

1. *Half-trained:* We train a 4-layer feed-forward neural network for only 50 epochs, resulting in around 50% increase in Mean Squared Error than a student trained for 400 epochs.
2. *Reversing Difficulty:* We train a 4-layer feed-forward neural network for 400 epochs, but with only 20% of the data containing reverse acceleration actions, to simulate difficulty with reversing.

Next, following the approach described in Sec. 3, we create a pool of 25 scenarios covering a diverse set of skills returned by our trained SKILLEXTRACTOR. We then collect rollouts from each synthetic student across all scenarios, use Alg. 2 to identify which 3 skills each synthetic student has least expertise in, and select drills from our offline drill dataset that target these skills. Using an equal dataset size, we compare and report average student reward across 15 different evaluation sets, where the student is fine-tuned on $(s, a)$ sampled from these individualized drills (**Ind. Drills**) vs. random drills (**Drills**) vs. entire $(s, a)$ trajectories from the original expert demonstrations (**Full Trajectory**).

Our results in Fig. 3 show that for the synthetic Half-trained IL Student, fine-tuning on data from both **Ind. Drills** and randomly selected **Drills** significantly outperforms the **Full Trajectory** setting ($p < .05$, Wilcoxon signed-rank test), while the difference between **Ind. Drills** and **Drills** is not significant. This may be due to a half-trained IL agent not having low "expertise" in any particular region, but more generally benefiting from training further on good quality data. Interestingly, for

---

[5]Note that the skills extracted are specific to operating a joystick/pen with a computer mouse.

[6]We select Balinese because, in comparison to other scripts available in the Omniglot dataset, the writing trajectories were consistent across users. Assisted teaching when there are multiple ways of performing a task is an important future direction.

the Student with Reverse Action Difficulty, **Ind. Drills** significantly outperforms *both* randomly selected **Drills** and **Full Trajectory** settings, but randomly selecting **Drills** slightly underperforms **Full Trajectory**, perhaps due to several drills not containing as many reverse actions as in the full-trajectory data. These results suggest that while drills can benefit both types of synthetic students in general, due to being composed of very common skill sequences, the synthetic student designed to lack a certain kind of "skill" is more likely to strongly benefit from individualization. This presents a promising signal that our CompILE-based SKILLEXTRACTOR learns skills that align with our intuitions for what skills are necessary for the PARKING task. However, our eventual goal is to teach *humans* motor control tasks, so we next test the usefulness of our approach with human students.

**Does assisted skill-based practice improve human learning outcomes?** Before constructing and individualizing drills, it is important to verify whether the skills returned by SKILLEX-TRACTOR—the building blocks of our approach—do indeed benefit student performance, as well as in comparison to simpler heuristics. For example, CompILE, as well as other skill discovery methods, take as input the expected number of skills $K$ per expert demonstration. A simpler **Time Heuristic** approach could instead be to assume that skills are temporally ordered and have equal length, and just split each expert demonstration into $K$ equally-sized segments. If such a heuristic sufficed, then

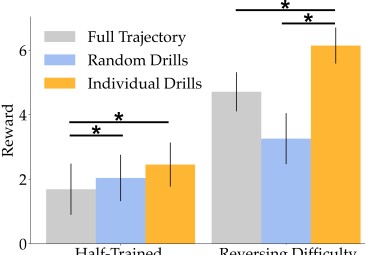

Figure 3: The effect of **Ind. Drills** and random **Drills** for synthetic students.

it may not be necessary to leverage more complex skill discovery algorithms. We therefore run user studies for both PARKING and WRITING environments comparing three different settings:

1. **Skills:** Each user practices trajectories corresponding to the 3 most common skills returned by SKILLEXTRACTOR based on CompILE parameterized with $K$, for 3 sessions each.
2. **Time Heuristic:** We temporally split each trajectory into $K$ intervals of equal length, and each user practices 3 different intervals for 3 sessions each.
3. **Full Trajectory:** Each user practices 3 different full trajectories of an expert demonstration (e.g. parking a car from the start state in PARKING, an entire character sequence in WRITING).

We set $K = 3$ for PARKING and $K = 8$ for WRITING. For fair comparison, the number of time-steps users are able to practice in the full trajectory and time-heuristic modes is roughly equivalent. Because our users may differ widely in prior experience (e.g. a video game player may be more familiar with the joystick in PARKING), we measure and report *Reward Improvement*, or the difference in average reward across 5 random evaluation scenarios and 2 random pre-test scenarios.

Our results in Fig. 4 show significantly ($p < .05$) higher reward improvement for students in both WRITING ($N = 25$) and PARKING ($N = 20$) in the **Skills** setting vs. **Full Trajectory** setting. This suggests that teaching students the necessary skills for a motor control task is extremely beneficial to improve learning. However, comparison between the **Skills** and **Time Heuristic** is interestingly inconsistent between environments, where **Time Heuristic** significantly

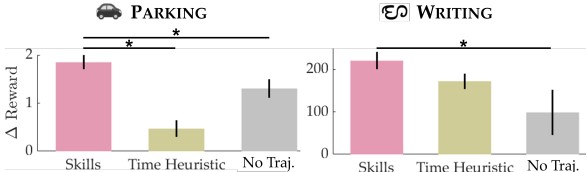

Figure 4: Students receive significantly higher reward improvement when practicing with the **Skills** setting than **Full Trajectory**. A simple **Time Heuristic** results in inconsistent performance across environments, demonstrating the necessity for a more reliable way to identifying skills for teaching.

leads to worse student learning outcomes in PARKING, but not for WRITING. One possible explanation might be that individual skills in WRITING, such as characters or certain curves, might be semantically intuitive on their own, and therefore even if **Time Heuristic** presents different characters as the same overall "skill", there is less confusion for users. On the other hand, because of the challenging dynamics in PARKING, a user may rely more on practicing skill trajectories that have clearly similar actions. Because heuristics are often environment-specific and complex, we view these results as an encouraging sign of the suitability of intelligent skill-discovery methods for teaching humans.

Finally, we ask all participants in a post-study survey *"What else would have been helpful for you to learn how to write the characters or how to park a car?"*. One user for the WRITING task who was assigned practice *without* skills responded *"Separating the characters"*. Responses from users who were assigned skill-based practice included *"Practicing a string of 2 characters"* and *"More time and more repetition"*. These responses motivate our next set of experiments on the effect of drills.

**Does practice with novel drill sequences outperform skill-based practice?** Though our previous results demonstrate skill-focused practice leads to stronger learning outcomes in both environments, we hypothesize it is equally important to learn how to combine skills that often co-occur via *drills*. For example, a qualitative look at the skills in Fig. 6 suggests some of the discovered skills consist of few actions, such as a short dot in WRITING or only a few time steps in PARKING. On their own these may not be useful skills to practice, but may be important to teach in the context of other skills.

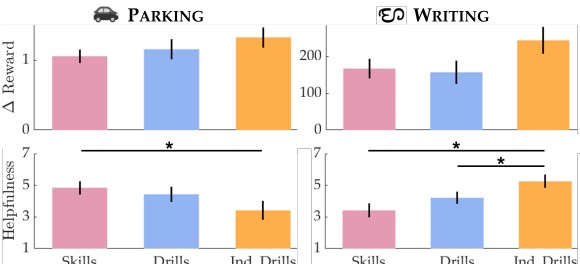

To compare with our individualization method, we create a fixed set of 5 pre-test scenarios covering a diverse set of skills returned by SKILLEXTRACTOR for both environments. We select 2 random drills, and compare the change in student reward when practicing with drills (**Drills**) vs. practicing the same underlying skill sequences used to composed the drills as separate practice sessions (**Skills**). Overall, as shown in Fig. 5, we find no statistically significant difference in reward improvement when students practice repetitive drill sequences composed of multiple skills (Blue bar) vs. skills in separation (Pink bar).

Figure 5: Students marginally improve reward more when practicing with **Ind. Drills** than **Skills** for both tasks. In WRITING, students significantly prefer **Ind. Drills** over both random **Drills** and **Skills**, yet this does not hold in PARKING, where targeted practice of reversing may frustrate students.

Possible explanations include the increased number of practice sessions in the **Skills** setting (because we keep the number of total time-steps equivalent across settings) mitigating benefits of repetition or practicing skill transitions in **Drills**. However, we next ask whether we may observe benefit from drills if we were to individualize them to students.

**Does individualizing drills improve student learning outcomes?** For each student, we run a CompILE-based SKILLEXTRACTOR immediately after receiving all pre-test student trajectories. Fig. 6 demonstrates the top 2 skills identified as "low expertise" across all students for both PARKING ($N = 20$) and WRITING ($N = 25$) environments. SKILLEXTRACTOR combined with Alg. 2 is indeed able to identify a wide set of low-expertise skills in student trajectories, ranging from more popular skills (e.g. "horn" shape in WRITING) to rarely difficult skills (e.g. steering the car forward in a straight line in PARKING). Importantly, this is despite the likely strong differences between human student and expert trajectories, emphasizing the viability of our approach to teaching humans.

Next, we select 2 unique drills targeting each identified skill, and provide them to students for practice before measuring reward improvement on an evaluation set. Fig. 5 shows individualized drills (**Ind. Drills**) lead to stronger reward improvements for students than **Skills** with marginal significance ($p \approx .06$ WRITING, $p \approx .09$ PARKING), supporting our hypothesis that individualizing drills leads to better learning outcomes. We visualize all student trajectories before and after practicing **Ind. Drills** in the Appendix.

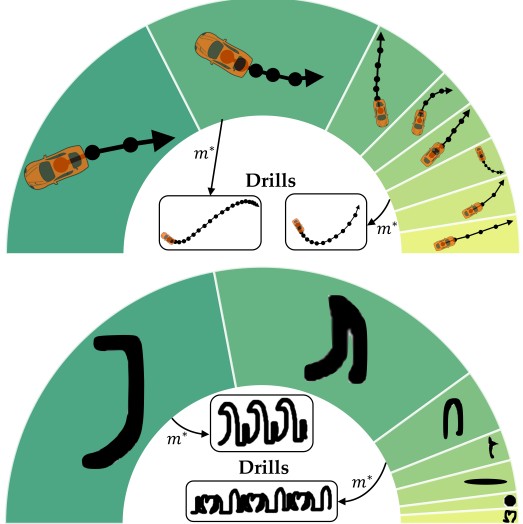

Finally, we ask students to rate the helpfulness of practice sessions from 1-7 (1 = least helpful), and show in Fig. 5 for WRITING, students strongly preferred practice sessions from **Ind. Drills** over **Drills** and **Skills**. However, despite the higher average reward improvement for **Ind. Drills** in PARKING, students who participated in **Skills** practice rated the helpfulness of their sessions significantly higher. Though there may exist many explanations for this result, including student preference for shorter sequences, we observe in evaluation rounds that users who received **Ind.**

Figure 6: Alg. 2 identifies a wide range of skills as low expertise across students for both environments. Reverse actions are identified as the most common skills for students to improve on in PARKING, while common curved-shapes are identified for WRITING.

**Drills** were around twice as likely to try **reversing** into a parking spot than other students (**53% vs. 27%**). This is likely due to our individualization method providing drills with reversing skills, which

were more common targeted skills (see Fig. 6). Crucially, in these scenarios *the expert action is also to reverse*, suggesting that while individualization may create a more frustrating learning experience, it may still help students achieve stronger learning outcomes by teaching more challenging skills.

## 6 Discussion

Our work is a first step towards building AI-assisted systems to more efficiently teach humans motor control tasks. However, there exist a few important limitations, which we now outline below.

### 6.1 Extracting "Human Teachable" Skills

A possible limitation of our work is whether skills extracted via SKILLEXTRACTOR are suitable for teaching human learners. While one benefit of our framework is its agnosticism to the *type* of the expert, we could imagine that using an automated RL-trained expert, for example, might result in teaching skills that are unsuitable for human learners. Addressing this requires stronger cognitive models of human motor learning in order to iterate different types of skill decomposition and optimize for some notion of "teachability." We could consider adapting existing methods used to model learning over time in other education domains (e.g. mathematics), such as Deep Knowledge Tracing ([34]) and Item Response Theory, which typically represent questions as discrete items or with simple features, and student responses as binary (correct or incorrect). Unfortunately, adapting such methods to handle the rich information in student trajectories is quite unstable, likely due to high dimension and scale (e.g. > 1K timesteps). Furthermore, student response variation for the same scenario is a lot higher for motor tasks than in other tasks such as in mathematics, where within a short period a student likely answers the same question the same way. Finally, such approaches require large amounts of student data (on the scale of thousands). Further discussion, as well as results demonstrating the unreliability of asking human experts to identify skills instead, is in Appendix Section E.

### 6.2 Accounting for Student & Expert Multimodality

One important aspect of many motor control tasks is multimodality - there may exist many optimal ways to complete the task. Although one can naively extend our approach to handle this by collecting demonstrations from a diverse set of experts and identifying which expert to "match" a student with, understanding the different control preferences present in a motor task is complex. For example, in writing, while different stroke orders for the same character clearly constitute different modes, there exist more subtle differences, such as the degree of rotation used, or the sharpness of the trajectory. These may occur due to physical preferences or conditions such as arthritis, but are hard to automatically distinguish as separate modes versus noise. Cleanly defining the distinct modes of a control task likely requires strong domain knowledge combined with large amounts of data across a diverse population. Furthermore, inferring which mode to teach a novice student is challenging, as pre-test trajectories only provide a limited amount of information. In traditional education settings, we often ask students questions such as whether one is left or right handed, and use that to inform how we teach. However, there may exist more obscure forms of preferences that require longer interaction with a student to elicit, and identifying such preferences is an important direction for future work.

### 6.3 Potential Failure Cases

Although we provide a general framework for assistive teaching of any arbitrary motor control task, including those that lack many expert instructors like teleoperating a novel surgical robot system, there are some settings that may be incompatible with our approach. These include situations where important skills do not appear in observed trajectories (e.g. the positioning of a hand on a ball before pitching in baseball), or there exists extreme stochasticity such that the same actions lead to drastically different outcomes, resulting in unintuitive dynamics for learners. Furthermore, in some tasks learning may not be feasible just from practice, but requires stronger forms of teacher-student interaction, such as physical guidance. We believe understanding the limits of demonstration-based teaching for motor tasks necessitate further insights from several areas within the broader NeurIPS research community, including cognitive science, reinforcement learning, and social aspects of machine learning.

## 7 Acknowledgements

We thank all anonymous reviewers for their valuable feedback. We acknowledge support from Point72, Ford, AFOSR, and NSF Awards #2218760, #2132847, and #2006388. Megha Srivastava was also supported by the NSF Graduate Research Fellowship Program under Grant No. DGE-1656518.

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
