**Note**: Additional visualizations of our experiments can be found here: 

## A    Broader Impact & Ethics Statement

AI-assisted teaching of motor control tasks can provide significant benefits such as more reliable teaching to individual students with different abilities (e.g. by leveraging more granular information about student actions), adaptability to any type of motor task or expert agent, and improved safety by reducing burden on human teachers for safety-critical tasks. However, we emphasize that our approach is solely meant to *assist* human teaching, as there exist many important aspects of human instruction that would be challenging to replace, including providing inspiration and motivation, in depth knowledge of human physical limitations, and an awareness of the broader context of a specific motor control task. Further risks of our approach, and avenues to address them, include:

- **Bias of the expert agent.** The suitability of the skills we use for teaching relies on how diverse the set of demonstrations from an expert is. For example, if a writing task only contained demonstrations from right-handed experts, certain action sequences that may be harmful for left-handed students' learning may be chosen as skills. Understanding how CompILE performs over a mix of expert types, and how to enable more complex adaptation to a specific student's needs at the skill-identification step itself are important future directions.
- **Over-reliance on the expert.** Our work currently assumes that in order to learn a task, the student should practice drills built from how an expert performs the task. However, a student should also be encouraged to learn when it may be appropriate to differ from the expert's actions if it helps the student learn better. This requires knowledge of how individual skills serve the ultimate task's goal (e.g. understanding why we first turn to enter a parking spot), which future work on incorporating interpretability methods and natural language instructions into our approach can address.
- **Student physical constraints during learning.** In many tasks, certain action sequences may physically be easier for an expert to perform than a student, and may perhaps even be dangerous for a student to practice without building up necessary techniques. This can be addressed by leveraging more complex hierarchical approaches to skill discovery (e.g. which skills should be mastered before attempting others) and incorporating knowledge of human physical constraints (e.g. degree of feasible wrist rotation). Another interesting direction for future work is to compare skills identified at different levels of expertise, and ascertain whether skills identified from a "medium-level student" may actually be easier, and less physically demanding, to *teach* with than those identified from an expert.

## B    Notation Glossary

For convenience, we provide a glossary of all mathematical notation used in our framework.

| Term | Meaning |
|---|---|
| $\Xi^s/\Xi^e$ | Student/Expert Scenarios |
| $\tau_\xi^s/\tau_\xi^e$ | Student/Expert Trajectories for scenario $\xi$ |
| $M_\xi^e$ | Set of skill labels corresponding to an Expert $e$'s trajectory for scenario $\xi$ |
| $E$ | Expertise vector for a given student |
| $b^\tau$ | Boundary of a skill subsequence in a trajectory $\tau$ corresponding to a particular timestep |
| $\tau_m^e$ | Segment of an expert $e$'s trajectory from interval $[b_{j-1}^{\tau^e}, b_j^{\tau^e})$ such that $m_j^{\tau^e} = m$ |
| $\oplus$ | Concatenation operator that stitches together action sequences when creating drills |

## C  Student Trajectories Before and After Practicing w/ Individualized Drills

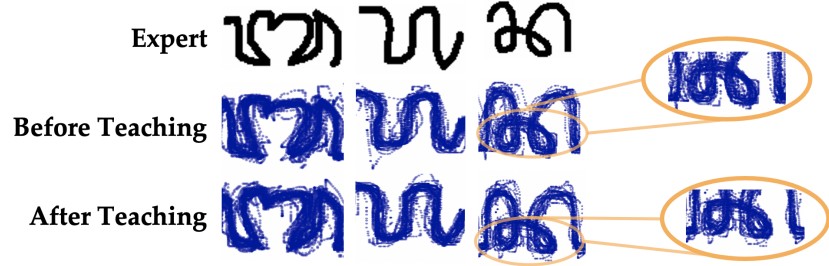

Figure 7: Overlay of all student trajectories of 3 Balinese characters for the WRITING task before teaching with individualized drills and after. For the rightmost character ("na" in Balinese), students learn to draw a smaller second loop at a lower angle, following the expert more closely. For other characters, we find that students exhibit less noise, and off-character strokes more closely following the original character.

We note that while Fig. 7 compares student trajectories before and after teaching for one particular set of Balinese characters, the reported values are averaged across all pre-test and evaluation rounds. Furthermore, common failure modes in the pre-test rounds before teaching that are not captured by this visualization include students giving up early and letting go of the mouse, and students re-tracing their characters.

## D  Comparison Between Student and Expert CompILE Outputs

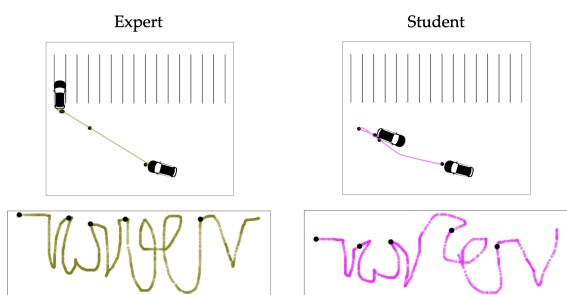

Figure 8: SKILLEXTRACTOR outputs for Expert (left) and Student (right) trajectories for both the Parking (top) and Writing (bottom) tasks. Black dots represent skill boundaries identified by CompILE.

Comparing SKILLEXTRACTOR output boundaries on trajectories from both our experts and students for both tasks, we can see that CompILE is able to segment both types of trajectories, but the noise in student trajectories leads to a failure of identifying the necessary skills for the task. We leverage this information to identify which skills students are struggling with. For example, in PARKING, the student is clearly unable to park the car, but the initial movement towards the upper left is segmented similarly to the expert, so the student will largely be penalized for later parts of the trajectory.

## E  Challenges in Human Expert Skill Identification

To address the challenges of extracting "human teachable" skills discussed above, one may consider using human experts as part of the SKILLEXTRACTOR function. However, the key idea behind our work is that people who are experts at performing a task may not be expert at teaching it, and may struggle to identify skills consistently.

To observe this clearly, we asked 3 different experts to annotate 10 successful trajectories of the PARKING task with boundaries corresponding to skills under unlimited time. The set of 10 trajectories only contained 3 unique demonstrations, allowing us to measure whether experts were consistent when providing skill annotations for the *same* expert demonstration. Fig. 9 shows that even for the

same trajectory, the same expert user may provide a completely different skill segmentation, and even identify a different number of skills. Even Expert 3, the most consistent expert across all duplicate trajectories, showed slight variation across trials - however, we note that CompILE's segmentations closely matched theirs. Moroever, by design, CompILE returns the exact same segmentations for the same action sequence. Overall, our user study results showed that human expert provided skills are often unreliable, leading to skill sequences with high variations that would be challenging to teach in a uniform way.

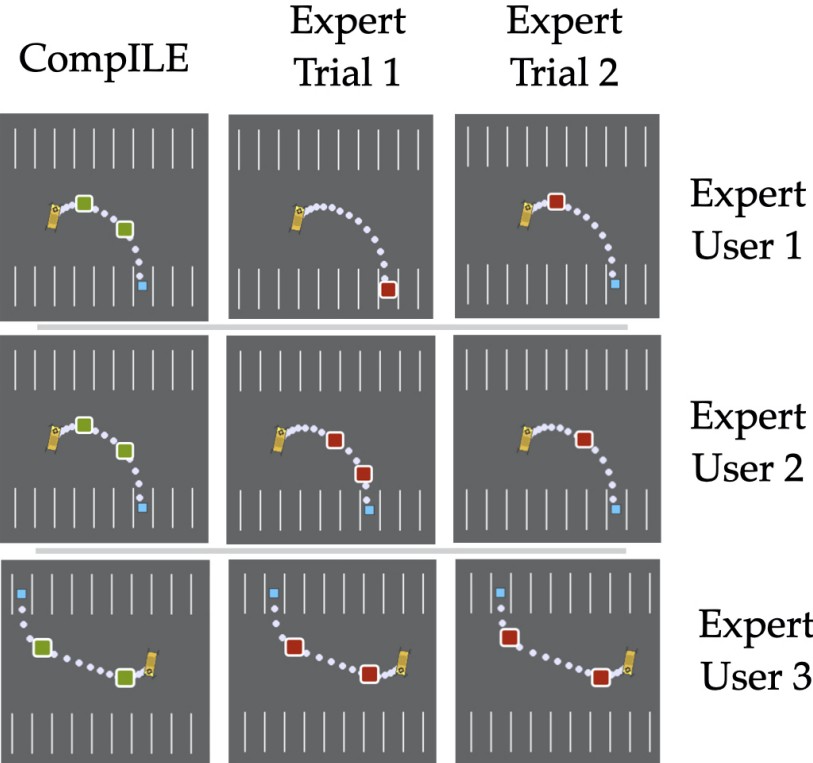

Figure 9: Skill segmentations from our CompILE-based SkillExtractor and 3 different human experts at the PARKING task. For a given trajectory, experts provided 2 segment boundary annotations (columns), with each row corresponding to a different expert. Expert Users 1 and 2 show high variability among their own segmentations, while Expert User 3 is more consistent and provides skill segmentations similar to the output of CompILE.

Finally, we note that a key aspect of our approach is scaling the ability to identify skills within a wide range of student trajectories for individualization. This is an even higher burden for human experts, who need to identify skills over trajectories that may widely differ from each other and the way the expert knows how to complete the task.

Therefore, in this work we attempted to incoporporate preliminary notions of "human teachability" when selecting between hyperparameter settings of our CompILE-based SKILLEXTRACTOR. Specifically, we filtered out skills corresponding to trajectories below a minimum length (due to human perceptual limits), and then chose the parameters that corresponded to the set of skills with highest entropy, with the intuition that a sufficiently diverse set of scenarios for a task would require a large variety of skills, and to minimize the risk of SKILLEXTRACTOR grouping two distinct skills as just one latent skill.

## F Impact of Training Time on Synthetic Student

Although we report results for both half-trained and reverse difficulty synthetic students after fine-tuning on 100 epochs, one natural question is the effect of training time. We examine this more closely with the "reversing difficulty" student, where Fig. 10 shows that as we increase the number of training epochs (equivalent to adjusting the $N_{rep}$ parameter in our IL setting) for a fixed set of 3

drills, student reward starts to plateau close to the average reward the expert receives. This shows that for our synthetic model, the largest learning gain occurs at the start of training.

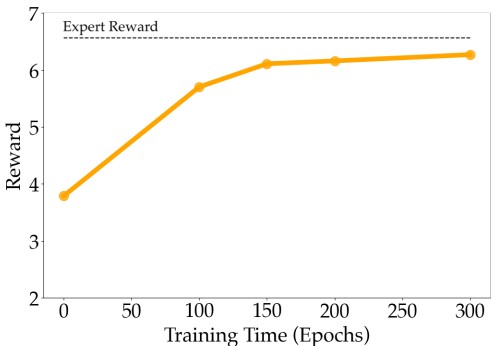

Figure 10: Reward starts to plateau over training iterations for the "reversing difficulty" synthetic student. Reported values are average reward over 100 random rollouts.

In practice, a teacher could also adjust the $N_{drills}$ parameter and repeat the entire teaching session by repeating the expertise identification method in Alg. 2, which may return different skills as a student improves over time. Planning the optimal overall curriculum is likely extremely task-dependent, and requires a much stronger model of student learning that incorporates concepts such as forgetting, which we leave for future work.

## G  Source Code

We provide all source code necessary to replicate our user study, for both PARKING and WRITING environments, as well as the trained CompILE models for both environments, at `https://github.com/Stanford-ILIAD/teaching`.

## H  Environment Pre-Processing

As described in Sec. 5 of the main paper, our PARKING environment is built off of the HighwayEnv goal-based task whereas our WRITING environment is custom built based off of the Omniglot dataset. For the purpose of simplifying our user study, we make the following modifications:

1. For PARKING, while we train our expert agent and skill-discovery algorithms across all possible parking goals, we only pick goals in the bottom right quadrant for teaching students in our user study as a simplification. To expand to all goals in the task, we believe further teaching time would be necessary due to the larger number and variety of skills required.
2. Likewise, for WRITING, we limit our sequences to contain only up to 5 different Balinese characters ("Na", "Ma", "Pa", "Ba", "Wa") to reduce the amount of skills required to learn the overall task.
3. Because crowdworkers in the Omniglot dataset differ in terms of interfaces used at the time of data collection, we "infill" all action sequences as a method of standardization. Specifically, we infill between any two consecutive states that differ by more than 1 pixel. Because these infilled trajectories are used to train our CompILE module for SKILLEXTRACTOR, we likewise infill all user trajectories collected in our user study.

## I  Hyperparameters & Training Details

Here, we describe all necessary hyperparameters to replicate training our (i) PARKING expert agent, (ii) skill-discovery CompILE modules for both PARKING and WRITING tasks, and (iii) synthetic students for PARKING. All models are trained on 1 NVIDIA TITAN RTX GPU, and the longest training time (for the expert PARKING agent) is roughly 5 hours.

1. **PARKING expert agent**: We train a StableBaselines3 implementation of Soft Actor-Critic for $10^6$ epochs with a learning rate of 0.001, which achieves a roughly 100% parking success rate.
2. **PARKING CompILE module**: We train a CompILE module (using the code from [28]) for 2000 iterations with a learning rate of 0.001, batch size of 100, latent dimension of 16 (i.e. 16 possible

skills), prior expected length of skill segments of 10, and prior number of segments per expert demonstration of 4. As described in the main paper, for PARKING, we add a penalty to the original loss function that is the MSE loss between the state differences between two consecutive states in the CompILE reconstruction and that in the training data.

3. **WRITING CompILE module**: We train a CompILE module for 80 iterations with a learning rate of 0.005, batch size of 50, latent dimension of 24 (i.e. 24 possible skills), prior expected length of skill segments of 250, and prior number of segments per expert demonstration of 8. For both PARKING and WRITING, we find it necessary to set the latent dimension high as many latent codes correspond to zero skill segments.

4. **PARKING synthetic "half-trained" student**: We train a 4-layer feed-forward neural network for 50 epochs with a learning rate of 0.0005 and batch size 256 via behavior cloning on rollouts from the expert agent, which receives an eval MSE loss of 0.049.

5. **PARKING synthetic "reversing difficulty" student**: We train a 4-layer feed-forward neural network for 400 epochs with a learning rate of 0.0005 and batch size 256 via behavior cloning on rollouts from the expert agent with only 20% of the data containing reverse acceleration actions (negative y-value), which receives an eval MSE loss of 0.021.

## J  User Study

We recruited users on Prolific, a crowdsourcing platform to conduct research studies, as part of an IRB-approved study (Protocol No. 49406 reviewed by Stanford University). We recruited up to 25 users for each setting in our user study for both WRITING and PARKING tasks. Overall, participants were paid an estimated wage of 15 dollars per hour, and took on average 20 minutes to complete the entire study, including reading instructions, learning the motor control task, and completing a post-task survey.

Each student was provided a link to an instructions page, where they provided a username to access the interfaces we built for both tasks. Each interface included step-by-step instructions on the side. As described in the main paper, students participated in a series of pre-test tries at the task, a sequence of practice sessions, and then an evaluation round. In the PARKING task, due to its difficulty, practice sessions consisted of both expert demonstrations (with joystick movement corresponding to expert actions) and student practice mode, while the WRITING task practice sessions consisted only of practice (of either full sequences, skills, or drills). Furthermore, to help guide students, we overlayed the state sequence of the target skill/drill/full-trajectory during demo and practice sessions for all settings in PARKING. Finally, for both motor control tasks we imposed a time limit on students for pre-test, evaluation, and practice sessions, proportional to the length of the sequence. For fair comparison, we ensure that the total number of allowed time-steps is roughly equivalent between settings we directly compare with each other. We include images of the instructions and example interfaces for both tasks below.

Finally, each student participant completed a post-task survey where they provided ratings for how helpful they found the practice sessions for learning the control task, information about whether students used a trackpad or computer mouse, as well as any feedback about the interface or task itself. We found no significant impact on performance from whether a students used a trackpad or computer mouse. We provide the complete list of survey questions asked below.

Overall, students found the PARKING task particularly challenging, often asking for more practice sessions, which many found helpful (e.g. *"It was interesting to give it a go and see how I improved in a short time."*, *"I could see that I was improving as the experiment went on"*). Meanwhile, students participating in the WRITING task students enjoyed the educational experience of learning a new script (e.g. *"I am interested in writing forms and would one day like to learn some unusual scripts."*, *"interesting learning to write another language"*), but wished to learn more about the characters' meaning, motivating further research in making automatically-discovered skills (which may not necessarily be characters) more interpretable to students (e.g. *"I would like to know what Balinese characters I'm tracing and their meaning"*).

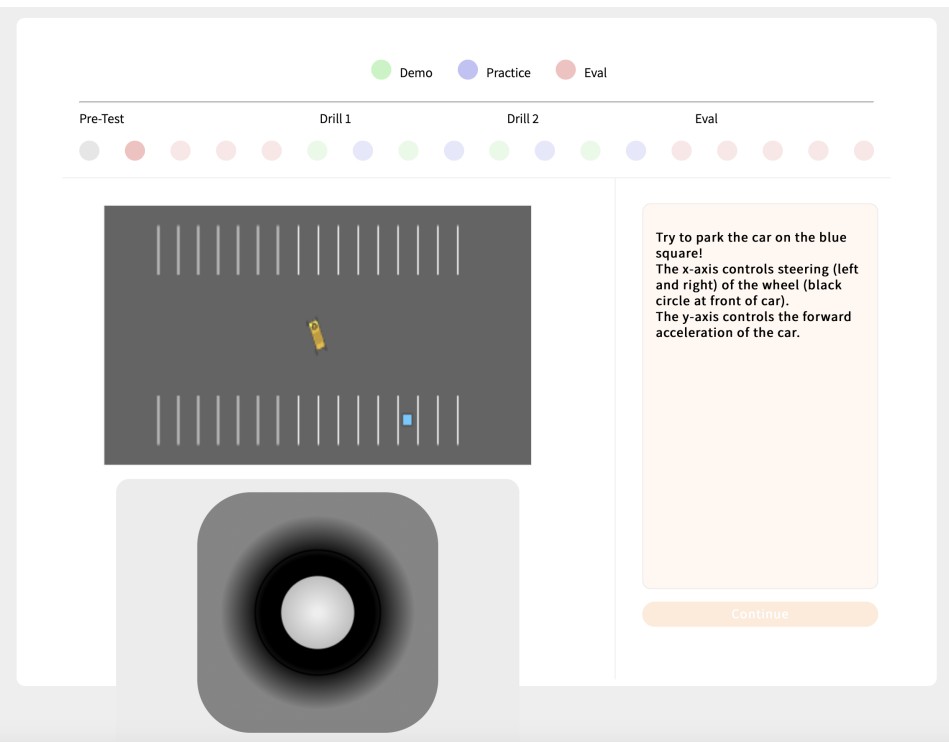

Figure 11: User study interface for the PARKING task where student participants learn how to park a car with a joystick controller. A black circle marks the front of the car, and the blue square marks the goal parking spot.

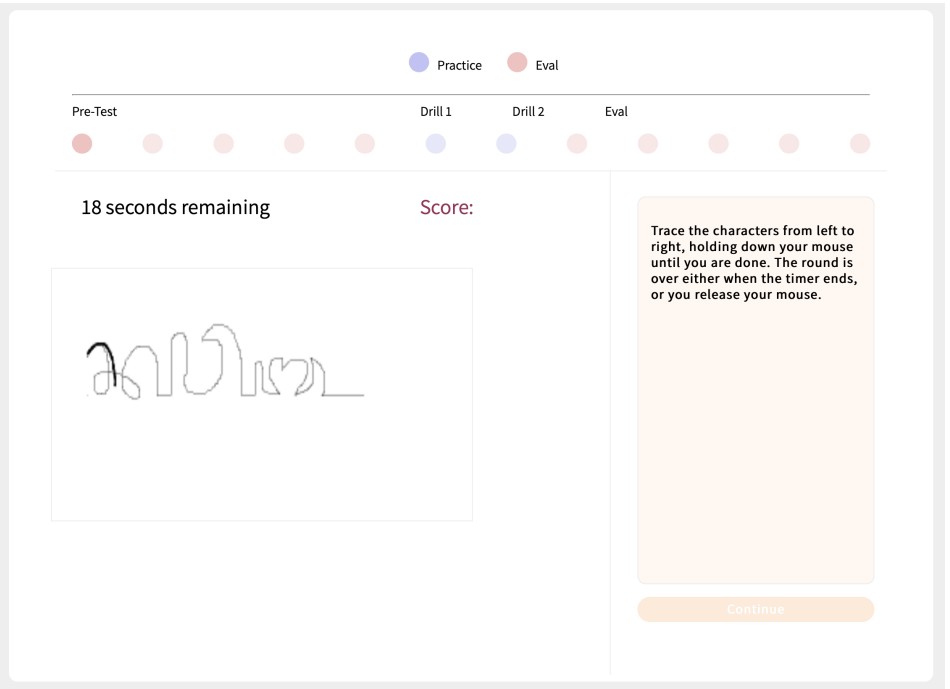

Figure 12: User study interface for the WRITING task where student participants learn how to trace Balinese characters. After the user lets go of their mouse, or when the timer is over, a score representing the reward would be displayed.

## Parking Control Experiment Instructions

Please carefully read the instructions below.

In this experiment, you will be playing a simple game where you try to park a car at a target parking spot, shown by a blue square. The computer knows how to perform this task, and at certain points, will provide you demonstrations of how to park the car and ask you to practice in order to help you improve at the task.

NOTE: You must be on a laptop or computer to participate. Additionally, it is important that you do **NOT** refresh or close the page during the experiment.

The game interface is shown below.

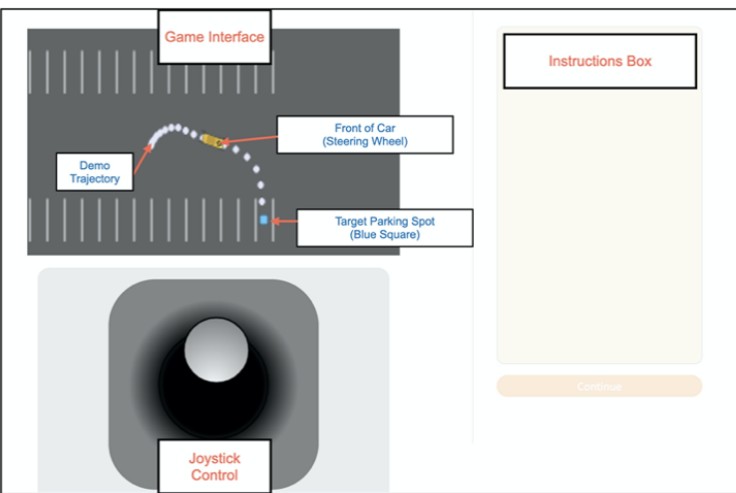

The goal of the game is to park the car in the spot marked by a blue square. To successfully park the car, you must stop the car over the blue square in a vertical orientation (parallel to parking lines). To control the car, you can use your mouse to control the joystick toggle on the interface. The x-axis controls steering (left and right) of the wheel (black circle at front of car). The y-axis controls the forward acceleration of the car.

The experiment consists of three phases.

**Phase I:** In Phase I, you will have a pre-test with two chances to try parking the car from start to finish. Use the joystick to control the car.

**Phase II:** In Phase II, you will practice 2 different drills needed to park the car. For each drills, you will alternate between watching a demonstration of the computer performing the drill (Drill Demonstration), and practicing that particular drill (Drill Practice). There are 2 drills in total. These drills are hard - try your best to use them to learn how to control the car! The following two paragraphs give more details on Phase II.

*— Drill Demonstrations.* During drill demonstrations, you will watch a video of how the computer completes that drill, which is repetition of a set of skills. To help you understand how the computer is controlling the car, the control joystick will move in the direction the computer chooses, and the vehicle will respond accordingly. The trajectory the car is following is shown in light gray. You should not touch any controls during these demonstrations.

*— Drill Practice.* During drill practice, you will have a chance to practice the same drill that was demonstrated to you by the computer. *To control the car, you should control the joystick on the interface with your computer mouse.* The joystick will move accordingly, as it did during the drill demonstrations. The trajectory the car should try to follow from the demo is shown in light gray, but it is okay if you cannot do this perfectly - try your best to use this opportunity to practice controlling the car!

**Phase III:** In Phase III, you will try to park the car from start to finish. You should try to apply the skills you learned in Phase II. Phase III has five rounds. Use the joystick to control the car.

Figure 13: Instructions for the PARKING task where student participants learn how to park a car with a joystick controller.

# Tracing Characters Experiment Instructions

Please carefully read the instructions below.

In this experiment, you will be playing a simple game where you try to learn how to write characters from the Balinese alphabet by tracing them from left to write. The computer knows how to perform this task, and will provide you different practice sequences in order to help you improve at the task.

NOTE: You must be on a laptop or computer to participate. Additionally, it is important that you do **NOT** refresh or close the page during the experiment.

The drawing panel in the game interface is shown below, with examples of high and low performing traces.

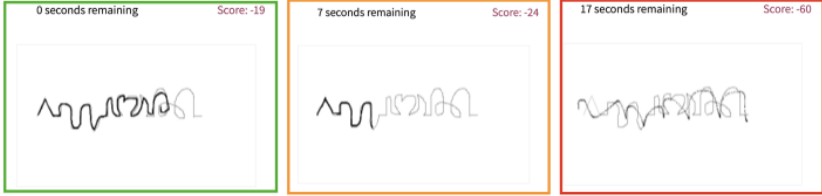

The goal of the game is to successfully trace, from left to right, the sequence of characters in the drawing panel, using your mouse as a pen. To successfully trace the sequence, you must trace over the entire sequence within the allotted time, while holding down your mouse/trackpad until done.
To control the pen, move your mouse to the start of the sequence and PRESS DOWN on your mouse to start drawing.
KEEP PRESSING DOWN your mouse as you trace over the entire sequence.
When you are done tracing, RELEASE your mouse.
The round is over when either you have released your mouse, or the timer is up.
You will receive a score after each round based on how well you traced over the sequence of Balinese characters.
Your score will be penalized if you either enter the whitespace too much and if you do not complete the sequence. *You should try to attempt tracing the entire sequence in the limited time.*

The experiment consists of three phases.

**Phase I:** In Phase I, you will have a pre-test with 5 chances to try writing over a sequence from start to finish. Use your mouse to control the pen.

**Phase II:** In Phase II, you will practice 2 different drills for writing Balinese characters. Each drill alternates between 2 skills to help you write Balinese characters. You will only get to practice each drill once. Use your mouse to control the pen.

**Phase III:** In Phase III, the evaluation round, you will try to trace a different set of sequences of characters. You should try to apply the skills you learned in Phase II, and get as high a score (closer to zero) as possible in each round.
PHASE III has five rounds. Use your mouse to control the pen.

Figure 14: Instructions for the SMALL CAPS WRITING task where student participants learn how to trace Balinese characters.

**User Study Survey Questions:**

1. Did you find the demos and practice sessions useful in learning how to park the car? / Did you find the practice sessions helpful in learning how to write the different characters? (Rating 1-7)
2. How easy was it to learn how to park the car? / How easy was it to learn how to write the characters? (Rating 1-7)
3. How easy was it to learn how to park the car? / How easy was it to learn how to write the characters? (Rating 1-7)
4. What else would have been helpful to learn how to park the car? / What else would have been helpful to learn how to write the characters?
5. What did you like about the experiment?
6. What would you wish to change about the experiment?
7. Did you use a laptop trackpad or a mouse to complete this study?