# OpenReview forum: "Assistive Teaching of Motor Control Tasks to Humans"
_NeurIPS.cc/2022/Conference — NeurIPS 2022 Accept_

### Official Review · Reviewer_W6Mp · 2022-07-06

**Rating:** 6
**Confidence:** 4
**Soundness:** 3 good
**Presentation:** 3 good
**Contribution:** 3 good

**Summary:**

The paper introduced an assistive teaching system for improving human skills on continuous control tasks. The core idea of the paper is to decompose a complex task into multiple reusable skills, assess the human skill level for each decomposed skill, and compose new personalized drills that help improve the weakest skills of the user. The system is evaluated on two tasks: parking a car with joystick control, and writing characters from the Balinese alphabet. The authors tested the proposed method on both synthetic learners as well as real human subjects in a user study. The experimental results shows that the method achieve improved learning performance in both cases for both tasks.

**Questions:**

1. During the evaluation phase, if a user finds a different sequence of operations from the expertise but still accomplishes the task (for example in the parking task there might be multiple, equally good solutions), would the proposed system penalize that and try to make the user follow the expert trajectory?

2. What do the learned skills look like for both tasks? It would be helpful to have some form of visualization for both expert trajectories and student trajectories to better understand the discovered skills and the evaluation process. In addition, for the writing task, can we use each letter as a skill?

3. In the synthetic student case, how many epochs were used during fine-tuning? Why are half-trained ones so much worse than the reverse difficulty ones even after fine-tuning?

4. In line 342, the Skills approach picks the 3 most common skills, while for writing 8 skills were discovered. Does this mean 5 of them are not seen by the user in this version of the method?



**Limitations:**

The authors have addressed some limitations of the work. One additional discussion point that would be helpful is some additional example problems where the proposed method would succeed and fail, which would not only provide insights on when the method is applicable, but also help better judge the potential impact of the work.

**Strengths And Weaknesses:**

Strengths:
1. The idea of leveraging skill discovery for teaching assistance is interesting and novel in my knowledge.
2. The proposed method that combines skill discovery, personalized evaluation and drill design, is reasonable and well elaborated.
3. The experiments support the proposed method in improving human learning performance.

Weaknesses:
1. One assumption that the method is making is that the decomposed skills can be acquired through demonstration, which circumvents the challenges of modeling human learning dynamics as described in the ‘What should the teacher do’ section instead of solving it directly. As such the method may not work for more complex tasks such as performing stunts where individual skills can also be difficult to acquire.
2. The demonstrated tasks do not require complex and reactive decision making, compared to something like a real-time strategy game. It’s not clear for more complex problems like those whether the skill discovery technique can still find reasonable low-level skills and whether the pipeline can scale accordingly.

---

> ### Author Response · Authors · 2022-08-02
> **Additional Discussion on Limitations**
>
> Thanks for your suggestion - we have added the following discussion of possible failure settings in Section E of our Appendix:
>
> *We provide a general framework for assistive teaching of arbitrary motor control tasks, including novel control tasks that may lack many expert instructors, such as teleoperating a novel surgical robotic system. The state / action representations, number of practice sessions, number of skills to practice, and types of drills created are all parameters of our method, and can be adapted for specific tasks. Furthermore, natural causes of stochasticity such as wind or body type can be naturally incorporated as part of scenario representations, if able to be measured.*
>
> *However, example tasks that may be unsuitable for our approach include situations where*
>
> *-important skills do not appear in the measured trajectory (e.g. the positioning of a hand on a ball)*
>
> *-human learning requires always having immediate overall task reward,  which focusing on sub-segments as skills will not provide*
>
> *-there exists extreme or adversarial stochasticity, so the same action sequence can lead to drastically different outcomes, making learning unintuitive for students trying to develop their own understanding of the task dynamics*
>
> *-skills that are important for learning but not required for optimal performance are not exhibited by expert demonstrations, and therefore not shown to the student*
>
> *- learning is not feasible just from student practice, but requires stronger forms of interaction between teacher-student, such as physical guidance and feedback*
>
> *- skills cannot be defined as solely action sequences, such as strategy-based skills. In some tasks, skills correspond to different action sequences depending on the state.*

---

> ### Author Response · Authors · 2022-08-02
> **Question 4: Most Common Skills**
>
>  Yes, recall our approach identifies the skills a student is struggling with most, and creates drills that target those skills specifically. However, note that since drills can consist of multiple skills stitched together (e.g. layup drills in basketball), while 3 different skills are targeted, the drills themselves may expose students to other skills. 8 refers to the overall set of skills identified as challenging across all users. Repeated iterations of our method, or increasing the N_drills parameter, would expose students to more skills.

---

> ### Author Response · Authors · 2022-08-02
> **Question 3: Synthetic Students**
>
> The half-trained student and reversing difficulty students were both equivalently fine-tuned for 100 more epochs during training, therefore trained on the same amount of data / number of gradient updates. The reason the half-trained student is much poorer in performance is that the reverse difficulty student only receives low reward for scenarios that require reversing, and even then can often recover by taking the longer, forward motion route (see Supplementary Videos linked in the Appendix). Meanwhile, the half-trained student has not learned optimal behavior across different scenarios.

---

> ### Author Response · Authors · 2022-08-02
> **Question 2: Learned Skills**
>
> Thanks for the suggestion! *We have added a new Figure 8 in the Appendix comparing CompILE-based skill segmentations of expert and student trajectories for both tasks.*  From our Writing task results, CompILE is able to identify shapes/strokes that occur in multiple letters, which may be more useful to practice than different letters as learning a shape skill helps with multiple characters.

---

> ### Author Response · Authors · 2022-08-02
> **Question 1: Multiple Optimal Solutions**
>
>  In our current version, yes, the user would be penalized and taught to follow the expert trajectory. One simple modification to address this would be to have the individualization method consider skill annotations from multiple CompILE modules trained on different possible experts, corresponding to different expert solutions, and then set the student penalties to be the minimum across different sets of skills.  However, inferring a priori which motor skills a student may prefer is complex as initial pre-test student trajectories may not be sufficient to cover the wide range of multi-modality for a particular task – therefore, we believe addressing this requires insights beyond the scope of a single work. We have added the following paragraph discussing this in Section D of the Appendix:
>
> *One important aspect of many motor control tasks is multimodality - there may exist many optimal ways to complete the task. Although our approach can be extended to handle this via collecting demonstrations from a diverse set of experts and identifying which expert to ``match" a student with, understanding the different modes present in a motor control task can be quite complex.*
>
> *For example, in a handwriting task, while different stroke-orders used for writing characters may clearly constitute different modes, there exist more subtle changes in action sequences, such as how the degree of motor rotation used, or whether the actions require sharp angles on a controller. These  may be due to strong physical preferences or conditions such as arthritis, but hard to distinguish as separate modes versus general motion trajectory variations. Cleanly defining the different modes of completing a task likely requires strong domain knowledge combined with a large amount of data across a diverse population.*
>
> *Furthermore, inferring what mode to teach a student who may be completely new to a control task is challenging as pre-test trajectories only provide a limited amount of information. In traditional teaching, we often ask questions such as whether a person is left or right handed, and use that to inform how we teach, but there may exist more obscure forms of preferences that require longer interaction with a student to elicit. Extending our work to a domain with such multimodality and the ability to engage with students over a longer period of time (which is tricky with crowdsourcing platforms) is an important direction for future work.*

---

> ### Author Response · Authors · 2022-08-02
> **Weakness 2: Complex Decision Making**
>
> Yes, we agree our demonstrated tasks do not exhibit complex decision making! This is actually intentional, as we would like to focus our work on challenges around teaching “motor control” tasks, which we believe prior work struggles to address. Specifically, the idea of practicing repetitive drills is particularly more common and useful for teaching motor control tasks, and differs from teaching high level decision making that might require demonstrating some type of optimal rules.  We primarily did not want to entangle what we believe are fundamentally different learning tasks for students. Furthermore, such task complexity would likely affect the number/length of our practice sessions and time required of users - important limitations for user studies with human subjects. It would be exciting to explore these more complex strategy games that combine motor control tasks with high level decision making in the future.

---

> ### Author Response · Authors · 2022-08-02
> **Weakness 1: Modeling Human Learning Dynamics**
>
> We agree that we are unable to fully capture human learning dynamics of motor tasks with our approach - this is quite a difficult problem foundational to understanding human intelligence itself, and current modeling approaches (e.g. Deep Knowledge Tracing) are unsuitable for us due to the complexity of trajectory data in motor control tasks.  However, we would like to emphasize that this is not an unreasonable limitation as human teachers also don’t have an accurate model of human learning, nor are we able to measure how inaccurate their model is,  yet they can effectively teach. Our method can potentially be applicable in even more complex settings: both the trajectory information and number of skills in 1 trajectory are hyperparameters of our method that can allow us to handle a variety of tasks – in the stunts example, how a human positions themselves beforehand and adjusts after landing can all be movement data incorporated in the trajectory. Furthermore, for quick stunts that consist of 1 “motion”,  creating a prior of 1 skill per trajectory will allow our method to default skills distinguishing between different scenarios (e.g. different starting speeds).
>
> Finally, we have added further discussion of the difficulties in modeling human dynamics in Section C of the Appendix, including:
>
> *Existing methods used to model student learning over time in standard education domains (e.g. mathematics), such as Deep Knowledge Tracing (\citep{piech2015deep}) and Item Response Theory, are currently unable to handle the high dimensionality, noise, and scale of motion trajectories inherent to motor control tasks. Furthermore, the variance of a student's trajectories of a same motor control scenario is a lot higher than in other education domains (e.g. within the same small time period, a student likely answers the same math question the same way), which may make modeling challenging. Additionally, such approaches would require large amounts of data (on the scale of > 1k students) for each different motor control task.*

---

> ### Author Response · Authors · 2022-08-07
> **Rebuttal Update**
>
> Dear reviewer,
>
> Given that the discussion period is soon ending, we were wondering if you had any follow-up questions that we could answer?  We believe that we clarified your doubts both in the updated manuscript and our rebuttal text, but any pointers to where our revision is lacking would be highly appreciated to improve our paper (changes are highlighted in purple).
>
> We would also be happy to engage in a discussion to address any other concerns that are currently affecting your review.
>
> Thank you for your time!

---

> > ### Comment · Reviewer_W6Mp · 2022-08-09
> > **Thanks for your response**
> >
> > The authors' response has addressed most of my questions. The additional discussions on the current limitations of the algorithm is greatly appreciated and would help readers better understand the scope of the work. Therefore I would maintain my current acceptance recommendation for the paper.

---

### Official Review · Reviewer_4JuP · 2022-07-10

**Rating:** 5
**Confidence:** 4
**Soundness:** 3 good
**Presentation:** 3 good
**Contribution:** 2 fair

**Summary:**

The paper presented an AI-assisted teaching algorithm that leverages skill discovery methods from reinforcement learning. The key idea of the proposed method is to decompose the skill to be taught into a few teachable sub-skills. By combining sub-skills into drill sequences, the algorithm then generates individualized curricula for different students.

**Questions:**

Questions:

1. I am not familiar with Balinese. However, I would like to ask: does the learned sub-skill have any semantic meaning? For example, in many writing systems, there are prescribed strokes or components in a character. Will CompILE be able to learn meaningful sub-skills in that case?

2. In the abstract, the paper mentions ‘individualize curricula to students with different capabilities’. What do the capabilities mean? Is it the current skill level of the student? In that case, is the training drill only generated based on the student’s initial performance? Is this caused by the lack of a human learning model?

3. In the supplementary materials, in figure 7, the performance improvement on writing before and after training is very subtle but the improvement over reward is significant, which is quite confusing to me.

Overall, I still think the paper is interesting and well-written. Therefore, I lean towards acceptance for now.

**Strengths And Weaknesses:**

Strengths:

1. The motivation of the paper is valid. I think the decomposition of a complex task into teachable skills is a simple and efficient strategy.

2. The paper is well-written.

3. The experiments are extensive. Both the PARKING and WRITING experiments are interesting.

4. Extending the skill decomposition in RL literature to human learning seems promising.

5. The code is provided for reproducibility.

Weaknesses:

1. My major concern is the discrepancy between the proposed approach and the target task (motor control). Particularly, I think the proposed approach does not capture the key challenge of teaching motor control skills. It seems that the success of the proposed method solely depends on human learning but ignores that humans may not be able to learn motor control skills without proper assistance. Let me explain. One of the characteristics of motor control skills is that it normally involves physical contact with the environment. In my opinion, to distinguish a teaching algorithm for a motor control task from the conceptual counterpart (e.g. learn classification), it is important to embed the notion of such physical interactions. For example, designing a practice sequence may not suffice in many rehabilitation tasks.  Without this, I don’t see a significant advancement over the existing Automatic Curriculum Learning (ACL) approaches made by the paper.

2. I think the paper lacks an analysis of why the learned teachable skills from reinforcement learning literature can be directly applied to human learning, which makes the result less convincing.

3. The paper investigates two tasks, PARKING, and WRITING. However, for both these tasks, the decomposition into teachable sub-skills can both be done by an expert. The approach will be more valuable if the author could show how the learned teachable skills compare with expert-defined teachable skills.

4. In this paper, the objective is to align students’ trajectory with expert demonstrations. From my perspective, this is not a good way to specify the learning objective, especially for complicated tasks with multiple optimal solutions.

---

> ### Author Response · Authors · 2022-08-02
> **Question 3: Figure 7 Performance Improvement**
>
> The reported reward improvement is an aggregate measure between all eval and pre-test examples, while Figure 7 just shows one particular example. Furthermore, the eval sequences are actually a different set from the pre-test sequences (but fixed across users), and reward measures pixel overlap between the student and true sequence. Thus it is primarily meaningful to compare reward improvement between different modes, as the scale could always be adjusted by a constant . Finally, one common failure for students in the initial pre-test rounds is running out of time and not completing the entire sequence in the time limit, leading to very low reward. We provide this additional clarification for Fig. 7 in the updated Appendix.

---

> > ### Comment · Reviewer_4JuP · 2022-08-08
> > **Thank you for the response.**
> >
> > Thank you for the detailed response and it is enjoyable to read. I really appreciate the authors' effort to clarify these points.
> >
> > In general, I am satisfied with the authors' response. I have a side comment regarding the human learning model. As mentioned by the authors:
> >
> > > While we agree with the importance of some limitations reviewers raised (e.g. lack of human motor learning model), we hope our rebuttal highlights why addressing these limitations constitute full, separate contributions themselves.
> >
> > Yes, I definitely agree that a good human motor learning model would serve as an individual contribution itself. However, I do not fully agree with the following statement
> >
> > > Existing methods used to model student learning over time in standard education domains (e.g. mathematics), such as Deep Knowledge Tracing (\citep{piech2015deep}) and Item Response Theory, are currently unable to handle the high dimensionality, noise, and scale of motion trajectories inherent to motor control tasks.
> >
> > Modeling human learning behavior in math in its own right is difficult. The success of existing methods in standard education domains relies on 1) A good representation of the human knowledge state (e.g. correctness of questions answered). 2) A large amount of human learning data that allows data-driven approaches to be deployed to learn the transition model. While they may not exist for motor skill learning now, it does not necessarily mean existing approaches (e.g. BKT, DKT, IRT) are not applicable for motor skill learning. It is important to draw the connection between these two domains.
> >
> > I will continue reading comments from other reviewers and adjust my overall evaluation accordingly. I see no reason to reject the paper for now.

---

> > > ### Author Response · Authors · 2022-08-09
> > > **Thank you for your response!**
> > >
> > > Thank you! Yes, we agree that it's important to better understand the connection between different education domains more, and work on adapting existing approaches such as DKT for motor skills learning -- while we faced difficulties initially working with DKT for control tasks, we definitely agree more should be explored before ruling out such approaches.
> > >
> > > We have modified that statement to the following instead:
> > >
> > > *For example, we could consider adapting existing methods used to model student learning over time in other education domains (e.g. mathematics), such as Deep Knowledge Tracing (\citep{piech2015deep}) and Item Response Theory, which typically represent questions as unique items (e.g. one hot vector encodings) or simple features, with binary student responses (correct/incorrect). These approaches can then be used to inform curricula design and relationships between different exercise scenarios, useful not only in traditional education domains, but also for motor control tasks (e.g. character strokes that have shared challenges in Writing). However, we found that adapting such methods to better incorporate the rich information contained in student motion trajectories led to unstable behavior, likely due to the high dimensionality, variance within scenarios, and scale (e.g. > 1k timesteps) inherent to motor control tasks.  Future work could consider leveraging our learned skills or learning how to better compress student trajectories, making such methods more tractable.*

---

> ### Author Response · Authors · 2022-08-02
> **Question 2: Individual Capabilities**
>
> Capabilities in our setting is on a skill-basis, and is indeed based on the pre-test performance of a student – it is not a generic “level”. For example, two students may have an equal success rate at a task, but have different types of weaknesses - our method for individualization would distinguish these two students and identify different sets of practice drills for them.
>
> This is indeed due to a lack of a human learning model. While there exist learning models for more standard education tasks, such as mathematics or language learning, such methods are very intractable when handling with motion trajectories.  We have added the following new paragraph in Section C of the Appendix further discussing limitations of current learning models:
>
> *Existing methods used to model student learning over time in standard education domains (e.g. mathematics), such as Deep Knowledge Tracing (\citep{piech2015deep}) and Item Response Theory, are  currently unable to handle the high dimensionality, noise, and scale of motion trajectories inherent to motor control tasks. Furthermore, the variance of a student's trajectories of a same motor control scenario is a lot higher than in other education domains (e.g. within the same small time period, a student likely answers the same math question the same way), which may make modeling challenging. Additionally, such approaches would require large amounts of data (on the scale of > 1k students) for each different motor control task.*

---

> ### Author Response · Authors · 2022-08-02
> **Question 1: Semantic Meaning of Skills**
>
> This is a great question! Note that CompILE identifies skills that are *reusable* for demonstrations across different scenarios, which may therefore be more meaningful in an “movement” sense than a semantic sense. Furthermore, for completely novel control tasks (e.g. operating an unfamiliar device), there may not exist strong existing notions of semantic meaning.
>
> However, for our specific Balinese writing task, we do see that the skills shown in Figure 6 tend to either correspond to entire characters (e.g. the “ba” consonant), or common shapes (e.g. the ending horn-shape in the “ka”, “ba”, “ma” constants). Strokes within Balinese characters do not carry semantic meaning, and Balinese in general tends to have single-stroke characters (judging by the Omniglot dataset annotations) – however, for another script, if semantically meaningful stroke was used independently across many different characters, it would likely be picked up as an individual skill.
>
> Aligning the learned skill trajectories with semantic descriptions that are more interpretable for human students is an exciting direction for future work.

---

> ### Author Response · Authors · 2022-08-02
> **Weakness 4: Align students’ trajectory with expert demonstrations / Multiple solutions**
>
> Yes, we agree that there is not a single way of achieving at performing a task, and that it is important to consider multiple optimal solutions for a task. One reason we may prefer aligning a student with an expert trajectory is because we may want a student to learn to perform a motor task “the safe way”, or in a way minimizing pain for their body, or in a manner more generalizable to other tasks. Simply learning to succeed at the task independently wouldn’t capture this, while teaching a student to “align” with an expert who has already identified a trajectory that captures these important characteristics would.
>
> One simple way our approach could be adapted to handle multiple optimal ways of solving a task would be to have the individualization method consider skill annotations from multiple CompILE modules trained on different experts, corresponding to different expert solutions, and then set the student penalties to be the minimum across different sets of skills.   However, inferring a priori which motor skills a student may prefer is complex as initial pre-test student trajectories may not be sufficient to cover the wide range of multi-modality for a particular task – therefore, we believe addressing this  requires insights beyond the scope of a single work. We have added the following paragraph discussing this in Section D of the Appendix:
>
> *One important aspect of many motor control tasks is multimodality - there may exist many optimal ways to complete the task. Although our approach can be extended to handle this via collecting demonstrations from a diverse set of experts and identifying which expert to ``match" a student with, understanding the different modes present in a motor control task can be quite complex.*
>
> *For example, in a handwriting task, while different stroke-orders used for writing characters may clearly constitute different modes, there exist more subtle changes in action sequences, such as how the degree of motor rotation used, or whether the actions require sharp angles on a controller. These  may be due to strong physical preferences or conditions such as arthiritis, but hard to distinguish as separate modes versus general motion trajectory variations. Cleanly defining the different modes of completing a task likely requires strong domain knowledge combined with a large amount of data across a diverse population.*
>
> *Furthermore, inferring what mode to teach a student who may be complete new to a control task is challenging as pre-test trajectories only provide a limited amount of information. In traditional teaching, we often ask questions such as whether a person is left or right handed, and use that to inform how we teach, but there may exist more obscure forms of preferences that require longer interaction with a student to elicit. Extending our work to a domain with such multimodality and the ability to engage with students over a longer period of time (which is tricky with crowdsourcing platforms) is an important direction for future work.*

---

> ### Author Response · Authors · 2022-08-02
> **Weakness 3: The decomposition into teachable sub-skills can both be done by an expert**
>
> Thank you for the suggestion! *We have added a new Figure 9 in the Appendix*, which includes a comparison of learned skills with human-expert-defined skills for the parking task, showing that skill identification by human experts can still lead to large inconsistencies. In particular, all three experts showed variations in their segmentations for the *same trajectory* across different trials, and 2 even varied in the number of skills they identified!  The CompILE outputs are not only exactly consistent across trials (by design), but similar to skills identified by the most consistent human expert.
>
> While these results are possibly unintuitive, in general human experts can struggle with developing curricula that identify skills about dynamics and motion, which can be harder to perceive and remember than for other domains. The difficulty of teaching athletes at the professional level, or the reason we still use specialized experts for tasks like driving, attests to the unique challenges of teaching motor control tasks.  Additionally, in rare tasks such as teleoperation of a new robotic system,  there may only be a very limited number of human experts available to identify skills. Furthermore, one strong benefit of using skill-discovery algorithms is that we can access *consistent* and *efficient* decomposition of skills, relying on the insight that it is a lot easier to find people who are experts at performing a task (e.g. car drivers) than it is to find people with the ability to identify fine-grained skills useful across scenarios..
>
> Consider parking, for example. While a human expert may be able to semantically describe some skills at a high-level (e.g. “right turn”), they may not be able to consistently identify motion trajectories corresponding  to this skill. Furthermore, without the ability to perceive individual action time-steps, they may be unable to dis-entangle fine-grained skills (e.g. varying degrees turn sharpness).
>
> Finally, our approach for individualization requires identifying what skills are present and missing from *student* trajectories. Therefore, it is not simply sufficient to have an expert simply identify skills generally for a task! Algorithms that learn to identify skills automatically can easily be used with student trajectories as inputs for our approach, while that would place a large burden on the human expert to manually annotate skills for each trajectory.

---

> ### Author Response · Authors · 2022-08-02
> **Weakness 2: Analysis of why learned teachable skills can be applied to human learning**
>
> Thanks for this suggestion – while CompILE and other approaches arose from the RL community, we believe the actual methods themselves are more general. For example, CompILE seeks to identify reusable action sequences across different demonstrations of a task, which is not only similar to other tasks like text segmentation in NLP, but also naturally in line with the way we define skills for human learning (such as learning strokes for writing).
>
> Furthermore, in Figure 4 of the main paper, we showed how the learned teachable skills not only outperform a time-based heuristic in learning improvement, but are also more consistent. This shows that how we segment a trajectory into skills *does* indeed influence human learning, and that our learned skills are more meaningful for human learning than other possible automatic approaches.
>
> Finally, we agree that there exists more room to improve the suitability of the learned skills. We attempt to incorporate preliminary notions of “human teachability” when selecting across hyperparameters of CompILE (e.g. by controlling for minimum skill length due to human perceptual limits). We expand on this further by adding the following new discussion in Section C of the Appendix:
>
> *Existing methods used to model student learning over time in standard education domains (e.g. mathematics), such as Deep Knowledge Tracing (\citep{piech2015deep}) and Item Response Theory, are  currently unable to handle the high dimensionality, noise, and scale of motion trajectories inherent to motor control tasks. Furthermore, the variance of a student's trajectories of a same motor control scenario is a lot higher than in other education domains (e.g. within the same small time period, a student likely answers the same math question the same way), which may make modeling challenging. Additionally, such approaches would require large amounts of data (on the scale of > 1k students) for each different motor control task.*
>
> *Despite these challenges, we attempted to incorporate preliminary notions of  ``human teachability" when selecting between hyperparameter settings  of our CompILE-based SkillExtractor. Specifically, we filtered out skills corresponding to trajectories below a minimum length (due to human perceptual limits), and then chose the parameters that corresponded to the set of skills with highest entropy, with the intuition that a sufficiently diverse set of scenarios for a task would require a large variety of skills, and to minimize the risk of \textsc{SkillExtractor} grouping two distinct skills as just one latent skill.*

---

> ### Author Response · Authors · 2022-08-02
> **Weakness 1: Discrepancy between the proposed approach and the target task**
>
> This is a great point for us to clarify more – we completely agree that motor control tasks are fundamentally different from other types of high-level discrete decision making tasks (e.g. deciding on what would be the best next chess move). However, we would like to argue that our approach and tasks actually account for physical interactions required by motor control tasks that go beyond human’s high level task plans, provided there is some mechanism to convey what physical interactions a user needs to practice. For example, consider the motor control task of playing the piano – there, drills are presented in the form of sheet music and notes. Information about the physical difficulties of the task (e.g. tricky hand movements) could be represented as part of the state or action using data from sensors, and thus expert / student trajectories would indeed (via pauses and variations) incorporate the physical challenges of performing the task. Thus, our generated curricula will capture skills that are physically challenging since intricacies of performing motor control tasks are present in the expert data. It is only necessary for students to map them onto the physical trajectories to practice.
>
> Under this framework, the way to view our experiments is learning parking/writing tasks under the specific form of physical input device control -- learning how to park a car using a mouse-controlled joystick, and learning how to write Balinese using a computer mouse. There, the difficulties with the physical contact aspect of the task (students letting go of the mouse after a period due to fatigue, limited rotation movement on the toggle, the relation between the time to hold the mouse and acceleration of the car), are reflected in the recorded actions of student trajectories, which in turn influence which physical actions they may need to perform again as practice.
>
> However, we agree that for some motor control tasks having feedback in the form of physical contact is important, such as the reviewer’s example of rehabilitation. For instance, visualizing the sequence of states and actions of a car — although sufficient in our task — might not be an applicable interface for demonstrating to others.  In such settings, our approach can still be useful as a form of identifying and creating practice sequences that target the particular aspects a student struggles with, and perhaps integrating information about the optimal way of performing the task via physical haptic feedback may aid with assistance. However, methods for incorporating physical feedback are  highly task dependent, and may require a separate set of analyses beyond the scope of our learning-focused contribution of generating useful practice sequences for these motor control tasks. Regardless, existing state-of-the-art approaches in automatic curriculum learning tend to represent educational tasks as items or simple features, unable to handle the high dimensionality of motion trajectories – our work goes beyond these approaches by leveraging skill-discovery methods,  thus better capturing issues specific to motor control.
>
> *We add further clarification of the scope of our motor control tasks (i.e. learning how to control the input device for Parking/Writing) to the paper, as well as a further discussion on settings our method may fail in the new Section E in the Appendix.*

---

> ### Author Response · Authors · 2022-08-07
> **Rebuttal Update**
>
> Dear reviewer,
>
> Given that the discussion period is quickly progressing we were wondering if you had any follow-up questions that we could answer?  We believe that we clarified your doubts both in the updated manuscript and our rebuttal text, but any pointers to where our revision is lacking would be highly appreciated to improve our paper (changes are highlighted in purple).
>
> We would also be happy to engage in a discussion to address any other concerns that are currently affecting your review.
>
> Thank you for your time!

---

### Official Review · Reviewer_XUpi · 2022-07-12

**Rating:** 6
**Confidence:** 4
**Soundness:** 3 good
**Presentation:** 3 good
**Contribution:** 3 good

**Summary:**

This paper presents an approach to providing training instruction for motor control tasks automatically by extracting skills from expert trajectories and combining them into drills based on user expertise. The authors use an existing framework for skill extraction and their main formal contribution is an approach for user evaluation and for individualized training curriculum construction. The paper evaluates different training scenarios with synthetic and real human users to analyze and demonstrate the benefits of their training curricula.

**Questions:**


In the formalism, one question that remains is related to the limitations introduced by the underlying deterministic model.
Why is a deterministic skill model sufficing ?

In the experiments, a few issues would be important:
Does the training curriculum yield optimal behavior eventually or does it lead to an intermediate “plateau” where presented skills and drills do not address areas where the user could further improve ?
How realistic was the underlying synthetic agent model to capture human skill acquisition ?



**Limitations:**

While the paper discusses some limitations and social considerations, some of the underlying assumptions and resulting limitations are not fully discussed. Among these are the deterministic skill model and the used synthetic agents for the first set of experiments.


**Strengths And Weaknesses:**

The application of AI to individualized training of human motor skills is an interesting application with many potential uses. The paper is well written and clearly presents the concepts and algorithms used. The experiments are intuitively presented and show important results for understanding of the effects on human learning.
The main weaknesses of the paper some potentially limiting assumptions that should be discussed more. For example, the target motor control task is defined as an MDP with deterministic transitions. Some discussion why deterministic transitions are sufficient here or what limitations this introduces would be useful.
Another issue is that in equation 5 the used indicator function II is not formally introduced.
In the experimental results, it would have been nice to also see some analysis of the effect of the amount of training on performance. In particular in the synthetic agent experiments, partially trained agents don’t seem to achieve high performance even after training. Would the proposed training approach actually get them to high performance or do they “plateau” ? In the human experiments, how close to the expert do users get ? This is hard to assess as no optimal performance reward estimate is provided.

---

> ### Author Response · Authors · 2022-08-02
> **How realistic was the underlying synthetic agent model to capture human skill acquisition?**
>
> We would like to emphasize that the main goal of synthetic agent models is not to match human learning, but instead as a medium for better evaluation: for instance synthetic models allow us to control for specific conditions such as creating agents with different capabilities (e.g. “reverse difficulty”) and this provides an approach for effectively evaluating our individualization approach.  More generally, designing accurate synthetic models of human motor learning is extremely challenging, requiring us to attempt to answer long standing questions in AI, i.e., understanding and abstracting human intelligence, adaptation, and learning.  We will discuss some of these challenges in the following paragraph we added to the Appendix Section C:
>
> *Existing methods used to model student learning over time in standard education domains (e.g. mathematics), such as Deep Knowledge Tracing (\citep{piech2015deep}) and Item Response Theory, are  currently unable to handle the high dimensionality, noise, and scale of motion trajectories inherent to motor control tasks. Furthermore, the variance of a student's trajectories of a same motor control scenario is a lot higher than in other education domains (e.g. within the same small time period, a student likely answers the same math question the same way), which may make modeling challenging. Additionally, such approaches would require large amounts of data (on the scale of > 1k students) for each different motor control task.*

---

> ### Author Response · Authors · 2022-08-02
> **How close to expert do human students get / No optimal performance reward estimate**
>
> We apologize that this was missing. We have updated Section 5 of the paper with the optimal performance reward. Note that in our main paper, we reported reward improvement for human experiments, as users have different prior familiarity with the task, so an expert would actually have 0 improvement. Furthermore, while in parking some of our human subjects do get close to the expert in terms of final reward, writing is trickier as our reward function measures pixel overlap between the student and true (expert) sequence, and it is quite difficult to achieve perfect overlap.

---

> ### Author Response · Authors · 2022-08-02
> **Indicator Function**
>
> Thanks for pointing this out. We have updated the paper to formally introduce this function!

---

> ### Author Response · Authors · 2022-08-02
> **Deterministic skills transitions**
>
> This is a great point! We will first discuss why our assumption is reasonable/can be handled by our approach for most motor tasks, and then where limitations exist.
>
> **Why this assumption is reasonable:** Stochasticity is mild in most real world motor-control tasks such that action sequences can still suffice as skills. For example, many machine control tasks, such as both our Parking and Writing tasks, the transitions are indeed deterministic given the actions from the input joystick / mouse control, and the only randomness appears in the starting state. Other tasks share this property, such as piano playing (where the actions are finger joint movements). In tasks such as playing tennis, stochasticity from sources such as wind can be addressed by incorporating such information into the state representation (e.g. from sensors), and if not addressed, contribute a modest amount of natural noise to the action sequence.
>
> **Why our approach can handle some stochasticity:** In our experiments, we find that our SkillExtractor is able to segment demonstrations that differ due to noise into the same skills. This is because the method essentially identifies reusable action sequences within a set of demonstrations of a task, so a large enough number of demonstrations would be able to tolerate the presence of noise. Indeed, even human teachers in tasks such as tennis are still able to define and cluster action sequences into skills (serving, volleying, back-hand), despite small variations in demonstrations of those skills.
>
> **Limitations:** However, if there exist extreme or adversarial amounts of stochasticity in the transition function (e.g. the tennis ball randomly teleports after an action sequence), this can make learning unintuitive for students, who may learn by relying on the predictability of repeated practicing of a skill, and may not understand the purpose of their actions. Ensuring that the student is aware of the forms of stochasticity in the control task is important, and we have included discussion about this point in a new Section E in the Appendix on proposed failure modes of our method.

---

> ### Author Response · Authors · 2022-08-02
> **Partially trained agents don’t seem to achieve high performance/Does the training curriculum yield optimal behavior eventually**
>
> There are a couple of points here with respect to model improvement vs. amount of training for the synthetic agents.
>
> First, our partially trained agents don’t achieve high performance because they start with quite low performance initially (negative reward) as they are non-optimal for all scenarios, whereas the reverse-difficulty student is non-optimal only for scenarios requiring reverse actions. We have added text to the main paper to better explain the reward function for Parking. Additionally, we have included a discussion on the challenges of designing strong synthetic models of human motor learning in a new Section C in the Appendix.
>
> Furthermore, we emphasize that there are a couple different components to our approach that would affect degree of learning: **N_rep** (number of repetition in drills, equivalent to “training time”/epochs), **N_drills** (number of unique drills, increasing results in more skills seen in a teaching session), and then **the number of *iterations* of our overall method**, where each iteration includes the individualization step, potentially identifying a new set of skills for the student to practice.
>
> For N_rep, as suggested by the reviewer, *we provide a new plot in Figure 10 of the Appendix* showing that as training time (e.g. N_rep) increases for the synthetic student, the student’s reward does indeed plateau, but at a value close to the mean expert reward. This additionally shows that for our synthetic student, the largest learning gain occurs at the beginning.
>
> In practice, a teacher could also adjust N_drills, exposing students to a more diverse set of skills, as well as the overall number of teaching session - for example, practicing with the 3 most difficult skills in the first session, followed by focused teaching on just the hardest 1 skill, and then general practice with all skills in the task. While increasing N_drills would naturally improve performance for the overall task, what is the optimal value and curriculum plan likely strongly varies between motor control tasks (e.g. learning to play a piano scale vs. learning how to perform a triple-axel require different time scales). The majority of prior work on *curriculum planning* focus on simpler domains like foreign language learning.  Adapting their methods to handle the high-dimensionality and variance inherent to motor control tasks, in order to best help a student achieve optimal performance,  is an exciting direction for future work.

---

> ### Author Response · Authors · 2022-08-07
> **Rebuttal Update**
>
> Dear reviewer,
>
> Given that the discussion period is soon ending, we were wondering if you had any follow-up questions that we could answer?  We believe that we addressed your concerns both in the updated manuscript and our rebuttal text, but any pointers to where our revision is lacking would be highly appreciated to improve our paper (changes are highlighted in purple).
>
> We would also be happy to engage in a discussion to address any other concerns that are currently affecting your review.
>
> Thank you for your time!

---

### Official Review · Reviewer_uRWk · 2022-07-14

**Rating:** 6
**Confidence:** 2
**Soundness:** 3 good
**Presentation:** 2 fair
**Contribution:** 4 excellent

**Summary:**


This paper proposes a framework that allows an AI agent to teach the human to do motor tasks, e.g., using a joystick for parking a car in simulation, or writing letters in an unfamiliar language. This framework's key advantage is that it will decompose the motor task into multiple skills, and compose those skills into individualized drills for the human student to practice.

The method works as follows:
1. Assuming that there are expert demonstrations of the given task. Then, the system first calls SKILLEXTRACTOR to extract skills from the demonstration. Each skill is represented as a sequence of actions and an interval where the skill is chosen to be executed.
2. Diverse scenario selection: choose a set of scenarios that can cover all skills via a greedy algorithm.
3. Identifying individual expertise in skills: ask the student to perform the task in those scenarios, and estimate her expertise.
4. Creating individualized drills: from step 3, identify the skills with low student expertise. Then, from the expert demonstration, identify N_drills n-grams involving the target skill with high frequency among all expert demonstrations.
Then, the student will practice each drill for N_rep times.

Results:
* Simulated student
  * Conclusion: Individualized drills (the proposed method) improve the total reward (statistically significant).
    * Compare among 3 methods
      * Ind.Drills: Individualized drills (the proposed method).
      * Drills: random drills.
      * FullTrajectory: the entire trajectory of expert demonstration.
    * Case 1: Simulated student as a half-trained behavior cloning agent
      * Ind.Drills > FullTrajectory's total reward (statistically significant).
      * Drills > FullTrajectory's total reward (statistically significant).
    * Case 2: Simulated student as a behavior cloning agent with reverse action difficulty
      * Ind.Drills > FullTrajectory's total reward (statistically significant).
      * Ind.Drills > Drills's total reward (statistically significant).

* Human student
  1. Conclusion: skill by SKILLEXTRACTOR improves the reward improvement (statistically significant).
    * Compare among 3 methods
      * Skills: SKILLEXTRACTOR (used by the proposed method).
      * TimeHeuristic: split the trajectory into skills evenly.
      * FullTrajectory: entire expert demonstration as a skill to practice.
    * Task 1: Parking
      * Skills > FullTrajectory's reward improvement (statistically significant).
      * Skills > TimeHeuristic's reward improvement (statistically significant).
    * Task 2: Writing
      * Skills > FullTrajectory's reward improvement (statistically significant).
  2. Conclusion: practicing with drills vs skills is not statistically significant.
    * Task 1: Parking
    * Task 2: Writing
  3. Conclusion: Individualized drills (the proposed method) improve the reward improvement (only marginal significance), and improves the subjective helpfulness (statistically significant)
      * Ind.Drills: Individualized drills (the proposed method).
      * Drills: random drills's reward improvement.
      * Skills: practice with skills's reward improvement.
    * Task 1: Parking:
      * Ind.Drills > Skills's reward improvement (only marginal significance)
    * Task 2: Writing
      * Ind.Drills > Skills's reward improvement (only marginal significance)

    * Task 1: Parking
      * Ind.Drills > Skills's helpfulness (statistically significant).
    * Task 2: Writing
      * Ind.Drills > Drills's helpfulness (statistically significant).
      * Ind.Drills > Skills's helpfulness (statistically significant).

**Questions:**


* Line 122: latent space M. Is the concept of latent space used in this paper? Latent space reminds me of VAE or graphical models. I was confused when seeing this in line 122.
* Line 314: does full trajectory mean that the student will just practice on the original expert demonstration?


**Limitations:**

N.A. All technical limitations are in `# Strengths And Weaknesses`.


**Strengths And Weaknesses:**


* Strength
  * Very interesting and novel problem formulation.
  * Online simulation-based user study in 2 tasks.

* Weakness
  * The proposed method assumes that the skill extracted by SKILLEXTRACTOR matches the skills that the student would need. However, in reality, each student could have a different set of skills extracted.
  * The writing could be easier to follow by reducing the notations, especially those notations with many super / sub-scripts.
  * User study with real human successfully shows that skill extracted by SKILLEXTRACTOR (used in the proposed method) improves the reward improvement with statistical significance. However, the user study failed to show that the proposed individualized drills improve the reward improvement (only statistically marginal significance).
  * Minor: Line 145: the paragraph about the POMDP approach could be shortened, because the paper later says that POMDP is intractable and proposes another approach. So the POMDP approach is not very relevant to the paper.
  * Minor: Fig.4: label "No Skills" should be "Full Trajectory"?

---

> ### Author Response · Authors · 2022-08-02
> **Weakness 2: Notation**
>
> Thanks for your suggestions. We have reduced the boundary subscripts in Section 3, and also provide a notation glossary in Section B the Appendix, as well as additional details on the skill segmentation procedure for increased clarity.

---

> ### Author Response · Authors · 2022-08-02
> **Weakness 3: Statistical Significance**
>
> We ran a power analysis using the empirical statistics of our already  collected data, and found thatthe results suggest we would need to collect around 200 more data-points to exhibit statistical significance with individualized drills, and 4,000 more for non-individualized drills - which is unfortunately out of our crowdsourcing budget, and we are thus limited by our small sample size.   We also believe that there is value in identifying approaches that lead to marginal improvement given our ultimate goal of helping human learners, and our results definitely demonstrate a trend of improvement even with a small sample size.
>
> Furthermore, under a one-sided t-test assumption, we do observe statistical significance (e.g. p≈0.03 for Writing), but we decided to not report this result for consistency and due to this additional assumption.

---

> > ### Comment · Reviewer_uRWk · 2022-08-10
> > **Response**
> >
> > Thank you so much for the detailed investigation!

---

> ### Author Response · Authors · 2022-08-02
> **Minor Weakness / Questions**
>
> **Minor Comments** Thanks for your suggestion. We have fixed the label and shortened the POMDP text! Our goal is to provide an overall framework for the teaching task that formalizes the problem while also providing a tractable solution. POMDP is simply a formal model for our teaching problem; however, the standard solutions for fully solving POMDPs are intractable. So instead, we propose a solution that leverages skills to address the tractability. We have shortened the text to make this more clear.
>
> **Question 1, Latent Space**: Yes, we use this formalism because many unsupervised skill discovery algorithms use auto-encoder frameworks, and aim to learn a good latent space of skills.
>
> **Question 2, Full Trajectory**: Yes, students practice the full trajectory. If this baseline performed well, it would mean we would not need to leverage AI-assistance for skill segmentation, although assistance may still be useful for individualization (identifying which full trajectories are best to practice).

---

> ### Author Response · Authors · 2022-08-02
> **Weakness 1: SkillExtractor Matching Student**
>
> Yes, we agree this is an important consideration!
> First, we would like to emphasize that our method utilizes a SkillExtractor (e.g. CompILE) to create  a library of possible skills from expert demonstrations that can be potentially useful for different students. So, each student might end up practicing different skills depending on what they uniquely struggle with.
>
> However, the reviewer might be pointing to the fact that there may not be a single optimal way of performing a motor control task, requiring our method to learn from a multi-modal set of experts. We emphasize that our overall method is agnostic to the choice of SkillExtractor. Moreover, our choice of CompILE is flexible enough to handle multi-modality in student learning behavior if provided a diverse enough expert dataset (covering all optimal ways to complete a task). One can simply adapt the individualization method to consider skill annotations from CompILE modules trained on different possible experts, and set the student penalties to be the minimum across different sets of skills.
>
> However, there are a number of challenges here that we believe are outside the scope of this work, including properly identifying multi-modality in motor control tasks and inferring a priori which motor skills a student has, as the student trajectories from the initial pre-test may not be sufficient to cover the wide range of multi-modality for a particular task. We have added the following paragraph discussing these challenges in Section D of the Appendix:
>
>
> *One important aspect of many motor control tasks is multimodality - there may exist many optimal ways to complete the task. Although our approach can be extended to handle this via collecting demonstrations from a diverse set of experts and identifying which expert to ``match" a student with, understanding the different modes present in a motor control task can be quite complex.*
>
> *For example, in a handwriting task, while different stroke-orders used for writing characters may clearly constitute different modes, there exist more subtle changes in action sequences, such as how the degree of motor rotation used, or whether the actions require sharp angles on a controller. These  may be due to strong physical preferences or conditions such as arthritis, but hard to distinguish as separate modes versus general motion trajectory variations. Cleanly defining the different modes of completing a task likely requires strong domain knowledge combined with a large amount of data across a diverse population.*
>
> *Furthermore, inferring what mode to teach a student who may be completely new to a control task is challenging as pre-test trajectories only provide a limited amount of information. In traditional teaching, we often ask questions such as whether a person is left or right handed, and use that to inform how we teach, but there may exist more obscure forms of preferences that require longer interaction with a student to elicit. Extending our work to a domain with such multimodality and the ability to engage with students over a longer period of time (which is tricky with crowdsourcing platforms) is an important direction for future work.*

---

> ### Author Response · Authors · 2022-08-07
> **Rebuttal Update**
>
> Dear reviewer,
>
> Given that the discussion period is soon ending, we were wondering if you had any follow-up questions that we could answer?  We believe that we addressed your concerns both in the updated manuscript and our rebuttal text, but any pointers to where our revision is lacking would be highly appreciated to improve our paper (changes are highlighted in purple).
>
> We would also be happy to engage in a discussion to address any other concerns that are currently affecting your review.
>
> Thank you for your time!

---

### Author Response · Authors · 2022-08-02
**Overall Response & Updated Paper Details**

We thank all reviewers for their detailed feedback, and are excited that reviewers found our work “very interesting” and “novel” (R1 + R4), showing “important results for understanding the effects on human learning” (R2) through “extensive” and “interesting” experiments, using a “simple and efficient” strategy (R3).

We have addressed all reviewer questions and concerns with both individual rebuttal comments and an updated paper. Concretely, our paper modifications include:
1. Added 3 new discussions in the Appendix describing settings where our proposed method could fail (R2, R3, R4), the suitability of extracted skills for teaching humans (R3), and handling multiple optimal solutions for a task (R1, R3)
2. Simplified notation in Section 3, added a notation glossary in the Appendix, and simplified discussion of the POMDP set-up (R1)
3. Added Figure 8 in the Appendix comparing boundary outputs from CompILE for both an expert and student demonstration for both Parking and Writing tasks (R4)
4. Added Figure 9 in the Appendix showing how human experts can be inconsistent when providing skills (R3)
5. Added Figure 10 in the Appendix demonstrating how the performance of synthetic students approaches that of the experts as training time increases  (R2)
6. Added further clarifying details for Figure 7 highlighting potential failure modes during pre-test rounds for users in the Writing task (R3)

While we agree with the importance of some limitations reviewers raised (e.g. lack of human motor learning model), we hope our rebuttal highlights why addressing these limitations constitute full, separate contributions themselves, due to requiring stronger insights from both cognitive and learning science areas that we hope this work can help motivate within the NeurIPS community.  We kindly ask the reviewers to let us know if there is any further clarification or information they would like us to provide.

---

### Meta-Review · Area_Chair_gzY7 · 2022-08-27

**Recommendation:** Accept
**Confidence:** Certain

**Metareview:**

The reviewers appreciated the extensive replies and the paper updates. Those managed to clear up most of the concerns of the reviewers that now all rate the paper cautiously positive. On the downside, now a lot of important material and discussion has been included in the appendix, while ideally the paper should be self-contained (i.e., no requiring the reader to also read the appendix).
The paper proposes a very interesting approach with a solid contribution and solid experiments. It is an important first step, while - like the authors point out - there are still quite a few limitations before this can be used practically, that will require separate papers to address.

**Award:**

No

---

### Decision · Program_Chairs · 2022-09-14

Accept